Reconstructing the dietary habits and trophic positions of the Longipterygidae (Aves: Enantiornithes) using neontological and comparative morphological methods

http://orcid.org/0000-0003-4310-3408 Clark Alexander D. 1 adclark@uchicago.edu
Hu Han 2
http://orcid.org/0000-0001-8244-6177 Benson Roger BJ 3
http://orcid.org/0000-0002-3898-8283 O’Connor Jingmai K. 4
1 Cincinnati Museum Center, Geier Collections & Research Center , Cincinnati, Ohio , United States
2 Department of Earth Sciences, University of Oxford , Oxford , United Kingdom
3 American Museum of Natural History , New York City, New York , United States
4 Field Museum of Natural History , Chicago, Illinois , United States
Hedrick Brandon
Electronic publication date: 2023 Mar 27
Publication date: 2023
Volume: 11
Electronic Location ID: e15139
Received 2022 Jun 2; Accepted 2023 Mar 7
Copyright: © 2023 Clark et al.
Copyright year: 2023
Copyright holder: Clark et al.
License: This is an open access article distributed under the terms of the Creative Commons Attribution License, which permits unrestricted use, distribution, reproduction and adaptation in any medium and for any purpose provided that it is properly attributed. For attribution, the original author(s), title, publication source (PeerJ) and either DOI or URL of the article must be cited.
License URL: https://creativecommons.org/licenses/by/4.0/

Keywords: Enantiornithines, Longipterygidae, Tooth morphology, Paleoecology, Paleoethology

Funding: European Union’s Horizon 2020 Research and Innovation Programme under the Marie Skłodowska-Curie grant agreement no. 101024572 This project has received funding from the European Union’s Horizon 2020 Research and Innovation programme under the Marie Skłodowska-Curie grant agreement No 101024572. The funders had no role in study design, data collection and analysis, decision to publish, or preparation of the manuscript.

==============================
The Longipterygidae are a unique clade among the enantiornithines in that they exhibit elongate rostra (≥60% total skull length) with dentition restricted to the distal tip of the rostrum, and pedal morphologies suited for an arboreal lifestyle (as in other enantiornithines). This suite of features has made interpretations of this group’s diet and ecology difficult to determine due to the lack of analogous taxa that exhibit similar morphologies together. Many extant bird groups exhibit rostral elongation, which is associated with several disparate ecologies and diets (e.g., aerial insectivory, piscivory, terrestrial carnivory). Thus, the presence of rostral elongation in the Longipterygidae only somewhat refines trophic predictions of this clade. Anatomical morphologies do not function singularly but as part of a whole and thus, any dietary or ecological hypothesis regarding this clade must also consider other features such as their unique dentition. The only extant group of dentulous volant tetrapods are the chiropterans, in which tooth morphology and enamel thickness vary depending upon food preference. Drawing inferences from both avian bill proportions and variations in the dental morphology of extinct and extant taxa, we provide quantitative data to support the hypothesis that the Longipterygidae were animalivorous, with greater support for insectivory.

Introduction

Morphologically and functionally similar features in extant taxa can provide data useful for constructing functional hypotheses and making niche predictions with regards to unique morphologies observed in extinct taxa (Clark & O’Connor, 2021). Examples in paleontological literature include inferring aquatic habits for hesperornithiforms based on morphological similarities to the avian orders Anseriformes, Gaviiformes, Suliformes, and Podicipediformes (Bell, Wu & Chiappe, 2019; Chinsamy, Martin & Dobson, 1998; Gingerich, 1973), reconstructing dietary habits of pterosaurs based on extant insectivorous and carnivorous dental morphologies (Bestwick et al., 2018; Clark & Hone, 2022; Ősi 2011), and determining forelimb function in drepanosauromorphs using similar features observed in extant arboreal and fossorial taxa to demonstrate functional correlations (Jenkins et al., 2020). However, predicting the ecology of extinct taxa can be difficult, particularly when attempting to attribute a specific niche based directly on comparison with extant organisms, among which a precise analogue may not exist (e.g., pterosaurs, drepanosauromorphs, and toothed birds of the Mesozoic).

It is tempting to compare Mesozoic bird species to taxa from among the great diversity of extant birds when seeking to determine possible behavior or ecologies. However, trophic comparisons are frustrated by the absence of teeth in all extant species, making inferences regarding both feeding behaviors and subsequent ecologies of toothed birds difficult when drawing from extant birds alone. Predicting diet entails and is limited to what the taxon might have eaten whereas predictions of ecology must incorporate predicted diet as well as hypothesize subsequent interactions between it and other taxa, the contextual paleoenvironment, and ultimately its position within the paleotrophic system reconstructed from co-occurring fossils. Differentiation between these two terms is often muddled in the literature.

The Longipterygidae are an unusual clade of Early Cretaceous enantiornithine birds, characterized by rostra that are proportionately more elongate ( ≥60% of total skull length) than other known enantiornithines (rostra ~50% skull length) and with teeth limited to the distal tip of the rostrum in both the upper (maxillary teeth absent) and lower jaws (O’Connor et al., 2009). Although possessing elongate rostra relative to other enantiornithines, compared to extant birds, these proportions are not technically longirostrine (rostra >70% skull length) but are mesorostrine (50–70%) (Marugán-Lobón & Buscalioni, 2003).

Known only from the Lower Cretaceous Jehol Group in northeastern China, the Longipterygidae is a relatively diverse clade consisting of at least five taxa divided into two subclades: the larger-bodied Longipteryginae consisting of Longipteryx chaoyangensis (Zhang et al., 2001) and Boluochia zhengi (O’Connor, Zhou & Xu, 2011; Zhou, 1995); and the smaller-bodied Longirostravinae, consisting of Longirostravis hani (Hou et al., 2004), Rapaxavis pani (Morschhauser et al., 2009), and Shanweiniao cooperorum (O’Connor et al., 2009). The Longipteryginae have large, labiolingually compressed teeth (i.e., blade-like) with crenulated apicodistal margins (Wang et al., 2015) and unusually thick enamel (Li et al., 2020) whereas the Longirostravinae have more gracile, peg-like teeth (enamel thickness currently unknown) (O’Connor, 2019). The large, crenulated, labiolingually compressed teeth of the Longipteryginae (Fig. 1) and the smaller peg-like teeth of the Longirostravinae (Fig. 2) likely reflect differences in diet and subsequently in their ecology. Several predictions regarding the diet and behavior of both groups have been put forth.

Figure 1 Longipteryx chaoyangensis (DNHM D2889).

(A) The skull of L. chaoyangensis exhibiting rostral elongation. (B) Artist reconstruction of the longipterygid L. chaoyangensis belonging to the subclade Longipteryginae. Illustration credited to and used with permission of Ville Sinkkonen. (C) Apicodistally recurved teeth restricted to the distal end of the rostrum (D) exhibiting crenulations. Crenulation features indicated by black arrows. Images of specimen credited to Jingmai O’Connor, used with permission.

Figure 2 Rapaxavis pani (DNHM D2522).

(A) The skull of Rapaxavis pani (DNHM D2522) exhibiting rostral elongation. (B) Artist reconstruction of a longipterygid, subclade Longirostravinae based on specimens of Longirostravis hani and R. pani. Illustration credited to and used with permission by Ville Sinkkonen. (C) Line drawing of the rostrum of R. pani, a longirostravine exhibiting small, conical teeth restricted to the distal end. Images of specimens credited to Jingmai O’Connor, used with permission.

Longipteryx chaoyangensis was originally predicted to have been piscivorous with a similar diet and ecology to extant piscivorous kingfishers (order Coraciiformes) based on similarities in skeletal morphology such as the elongate rostra, “powerful wings”, perching structure of the foot, and “relatively short hindlimbs” (Zhang et al., 2001). Some authors support this inference based on the unusually thick tooth enamel of L. chaoyangensis, which is reminiscent of some ichthyosaurs (Li et al., 2020), although tooth enamel is very thin in other dentulous, presumably piscivorous, bird genera (e.g., Ichthyornis and Hesperornis) (Dumont et al., 2016; Benito et al., 2022). Currently, there is no direct evidence for piscivory in the Longipteryginae, such as fossilized in-situ ingested fish. Other authors interpret L. chaoyangensis as hypercarnivorous relative to other enantiornithines based on the presence of enamel crenulations and the labiolingually compressed and recurved shape of the teeth, with a hypothetical diet consisting of insects and small vertebrates (Wang et al., 2015).

Members of the Longirostravinae were first predicted as mud-probers (Hou et al., 2004; O’Connor, 2019) and later reinterpreted as bark-probers (Morschhauser et al., 2009), though sediment probing hypotheses persist (Miller et al., 2022). In the initial description of L. hani, Hou et al. (2004) hypothesized that the delicate, elongated rostrum implied probing for prey within soft substrates such as mud, possibly in search of invertebrates, similar to members of the extant families Haematopodidae and Scolopacidae (O’Connor & Chiappe, 2011; Winkler, Billerman & Lovette, 2020a). However, inconsistences in pedal morphology with extant soft sediment probers led other authors to suggest the elongate, delicate rostrum in L. hani and R. pani may instead have been used to probe under bark, similar to extant members of the avian families Certhiidae and probing genera within Furnariidae (e.g., Lepidocolaptes, Campylorhamphus, Xiphorhynchus), an interpretation more in line with the arboreal foot morphology in these taxa (Morschhauser et al., 2009; O’Connor, 2019; Wang et al., 2013).

Recently, a study attempted to use body mass, claw curvature, jaw mechanics, and finite element analyses to predict longipterygid diet (Miller et al., 2022). The authors conclude that longipterygids were likely insectivorous or generalists due ultimately to these diets covering the most functional morphospace in their analysis. Additionally, they concluded that pedal manipulation of prey items might have been similar to some extant birds of prey. However, the ability to wrap the pedes around possible prey items as justification for interpretations regarding grasping strategies is an inherently weak point given that, as stated by the authors, passerines (i.e., birds characterized by their ability to perch using the hallux) cover nearly all morphospace in their analysis.

Currently there is no well-supported hypothesis regarding the diet and ecology of the Longipterygidae utilizing quantitative data and comparative morphology that incorporates extant taxa beyond birds. Chiroptera, as the only extant group of toothed animals that utilize powered flight (Gunnell & Simmons, 2012), can provide insight useful for investigating how teeth primarily used for prey acquisition (i.e., canines) may have functioned in toothed birds. The order Chiroptera is divided into two suborders; Yinpterochiroptera (containing all previously known megabats and the microbat families of Rhinopomatidae, Rhinolophidae, Hipposideridae, Craseonycteridae, and Megadermatidae), and Yangochiroptera (containing the remaining families of the once Microchiroptera) (Hawkins et al., 2019; Teeling et al., 2005; Tsagkogeorga et al., 2013). Though extant chiropterans have heterodont dentitions, whereas longipterygids are considered to have homodont dentitions, we hypothesize that chiropteran canines, with their known primary function of piercing, puncturing, and crack propagation, and their distal location in the rostrum, make them suitable analogues for providing data to aid in dietary and ecological predictions for the Longipterygidae. We revisit longipterygid enantiornithines with regards to possible dietary functions of their unique rostral morphology and dentition drawing from extant taxa in order to provide enhanced, quantitative and qualitative predictions for both diet and possible behavior of this group of enantiornithines within the Jehol paleoenvironment.

Materials and Methods

Tooth enamel among extinct taxa

To assess enamel thickness values in a variety of extant and extinct taxa, ranging from ~0.009 to ~2,500 kg in mass, we gathered data from the published literature (Benson et al., 2014; Dumont, 1995; Dumont et al., 2016; Hwang, 2011 and Li et al., 2020; Mackiewicz et al., 2010; Selig et al., 2020; Serrano & Chiappe, 2017). Published data concerning enamel thickness in theropods (both avian and non-avian) is limited. In total, enamel thickness values were recovered for 24 mammals, 10 (non-dinosaurian) reptiles, seven non-avian theropods, and eight avian theropods (Table S1). Estimated mass and probable diet according to the literature were included for comparison. If a mass range was given, average mass was recorded. If species mass data was missing, the closest related taxon of similar size with mass information was used.

Dietary regime terminology

Unless specified elsewhere, we define carnivory as a diet that consists primarily of terrestrial vertebrate prey, piscivory as a diet that consists primarily of aquatic vertebrate prey, invertivory as a diet that consists of either aquatic, shoreline, or riparian-associated invertebrate prey (e.g., mollusks, arthropods, gastropods), insectivory as a diet that consists primarily of terrestrial invertebrates (e.g., insects), herbivory as a diet that consists primarily of seeds (granivory), fruit (frugivory), or foliage (folivory), nectivory as a diet that primarily relies on flower nectar, and omnivory as a diet that consists of a combination of animals and plants, with either making up the majority of the daily intake, possibly depending upon the season. Primary diets were sourced from Wilson, Mittermeier & Cavallini (2009), Billerman et al. (2022), and the University of Michigan Museum of Zoology’s animal diversity database (Myers et al., 2022). It is important to note that the taxa used within analyses all have primary food and prey preferences, though other resources also supplement their diets. An example is nectivory seen in primarily insectivorous species of birds (e.g., Nectariniidae). Each species is represented by their primary diet, though this often does not encompass the full range of their diet.

Geometric morphometric analysis

The fossilized teeth of the sampled longipterygids are well-preserved in lateral view, and thus allow for two-dimensional (2D) morphometric comparisons with extinct and extant taxa. All mammal and sauropsid skull specimens, as well as dentition photos, were obtained from the Carnegie Museum of Natural History (Pittsburgh, PA, USA). Using 2D geometric morphometrics (GM) we compared the tooth morphologies in L. chaoyangensis (Longipteryginae) and L. hani (Longirostravinae) with flying, gliding, and terrestrial extant species from Sauropsida (three orders, 17 families, 32 species) and Mammalia (six orders, 24 families, 80 species) (refer to Table S2 for included taxa). We also included eight additional extinct taxa including non-avian theropods and a non-longipterygid enantiornithine. Photo cataloging took place in tpsUtil64 (version 1.81) (Rolhf, 2021a) and landmarking in tpsDIG2 (version 2.32) (Rolhf, 2021b). Some photographed specimens that were too small to produce images of sufficient resolution to negate landmarking point overlap were digitally outlined and converted into vectorized shapes in order to properly landmark without overlap. Three landmarks and two curves were placed on each photo/vectorized shape of either the canines (e.g., chiropterans, canids), mesial incisors (e.g., rodents), or maxillary teeth (e.g., sauropsids) in lateral view (Fig. S1 and Table S3). Both curves were placed along the distal and mesial margins (Fig. S1). More distally-located teeth in the jaw were chosen due to their primary function in initial contact with food items (e.g., animals or plant matter), crack propagation, and puncturing actions (Freeman, 1984; Freeman, 1988; Van Valkenburgh, 1988). Dentition-specific directional anatomy followed Hendrickx, Mateus & Araújo (2015). Body mass information for each species included was gained from Myers et al. (2022). Sources of measurements are also presented in Table S2). Five dietary categories were assigned to all the extant samples according to Wilson, Mittermeier & Cavallini (2009), Billerman et al. (2022), and the University of Michigan Museum of Zoology’s animal diversity database (Myers et al., 2022): carnivorous, insectivorous, piscivorous, herbivorous, and omnivorous as binary categorical variables. A subset of the previously listed dietary regimes (full list above) was used due to a lack of nectivores or riparian/shoreline invertivores present in extant dentulous taxa.

All of the digital landmarks were imported into R where semilandmarks were identified (version 4.1.1). A Generalized Procrustes Analysis (GPA) was performed using the gpagen() function from the R package “geomorph” (version 4.0.5), followed by a principal components analysis (PCA) using gm.prcomp() from “geomorph” (Adams et al., 2016). The significance of the shape changes among the five dietary categories was evaluated by performing a Procrustes ANOVA and phylogenetic Procrustes ANOVA using the functions procD.lm() and procD.pgls() from “geomorph” (Adams et al., 2016). Both lm and pgls analyses were conducted to reveal a more comprehensive understanding of the tooth shape variations among different dietary categories. The shape variations of the targeted tooth along different PC axes were visualized using plotRefToTarget() from “geomorph” (Adams et al., 2016). Euclidian distances within morphospace of taxa were measured to assess similarities between extinct and extant taxa tooth morphology.

Maximum clade credibility trees for Sauropsida and Mammalia were separately constructed from two sets of 100 molecular trees downloaded from https://vertlife.org/data/ (Upham, Esselstyn & Jetz, 2019; Tonini et al., 2016), using maxCladeCred() from the R package “phangorn” (V 2.11.1) (Schliep, 2011). These two maximum clade credibility trees were combined together to be used in the subsequent phylogenetic comparative analyses, using bind.tip() from the R package “phytools” (V 1.2) (Revell, 2012).

Bird rostral proportions

To examine associations between proportional and absolute rostral size and dietary regime associations, we sampled a variety of extant birds (156 families, 352 species). Measurements of total bird skull and rostral lengths taken in person were obtained from extant bird specimens housed in the collections of the Cincinnati Museum Center (Cincinnati, OH, USA), Carnegie Natural History Museum (Pittsburgh, PA, USA), and the Field Museum of Natural History (Chicago, IL, USA) (Table S4). Skull measurements were taken to the nearest 0.1 mm using digital calipers (OriginCal1P54). All other specimens were obtained from point-to-point Euclidean distances of 3D landmarks placed on digital models of museum specimens derived from CT scans (Bjarnason & Benson, 2021; Navalón et al., 2022; see Table S4 for list and links to original data on www.morphosource.org). Sexual dimorphism was not considered for rostral measurements as this feature does not significantly vary between sexes of extant birds (with few exceptions such as Phoeniculus purpureus which was incorporated in our data). Skull length was measured from the rostral-most tip of the bill (regardless of rostral curvature) to the caudal-most margin of the parietal. Rostral length was recorded from the rostral-most tip of the bill to the caudal margin of the lacrimal (i.e., the rostral margin of the orbit). When able, two specimens of a species were used in order to account for possible variation among individuals and possible damage to the rostral tip (51 species). Species with missing or completely missing rostral tips were not included. Body mass information for each species included was sourced from the Bird of the World online database (Billerman et al., 2022), which contains mass ranges for almost all known bird species. When a single body mass was given it was used for both minimum and maximum body masses for a species. Mass differences between sexes were present, but regardless, minimum and maximum for the species was used. If mass was not recorded, the next closest related species of similar size was used as a substitute. Scatterplots were created in R, using the program ggplot (Wickham, 2016).

The significance of the changes of the rostrum lengths among the seven dietary categories (i.e., carnivory, herbivory, insectivory, invertivory, nectivory, omnivory, and piscivory) were evaluated by performing phylogenetic generalized least squares (pGLS) regression of the form “rostral length ~ skull length + dietary trait”, using functions from the R package caper (Orme et al., 2013). In these models the term ‘dietary trait’ refers to any one of our diet items as a binary categorical variable (e.g., piscivore | not piscivore). This evaluates the independent effects of our dietary traits on variation in rostral length after accounting for skull length variation (i.e., it examines the dietary correlates of variation in rostral length among birds). We also evaluated the ecological signal of skull length, an overall size measure, using models of the form “skull length ~ dietary trait” using pGLS.

Similar to our GM analysis, a maximum clade credibility trees for extant samples included in the bird rostral analysis were constructed from a set of 100 molecular trees downloaded from https://vertlife.org/data/ (Jetz et al., 2012), using maxCladeCred() from the R package “phangorn” (Schliep, 2011).

Results

Tooth enamel among extinct taxa

Compared to other Cretaceous birds, L. chaoyangensis exhibits unusually thick tooth enamel, measuring 50 microns near the base of the crown, which exceed values for other toothed-birds such as Sapeornis chaoyangensis (21 microns), Jeholornis prima (seven microns), an indeterminate enantiornithine (6.2 microns), and an unidentified ornithuromorph (6.5 microns) (Li et al., 2020) (Table S1 and Fig. S2). Enamel thickness in L. chaoyangensis is also much greater than in the larger stemward paravians Anchiornis huxleyi (possible troodontid or early avian; 14.3 microns) and a microraptorine dromaeosaurid (7.1–11.1 microns), both of which were predicted to be animalivorous (Hone et al., 2022; O’Connor, 2019; O’Connor, Zhou & Zhang, 2011; Li et al., 2020; Xing et al., 2013; Zheng et al., 2018) (Table S1). Animalivory entails a diet comprised almost equally of both vertivorous (carnivory) and invertivorous (insectivory/invertivory) prey. In the more derived Late Cretaceous piscivorous ornithurine birds Hesperornis regalis and Ichthyornis dispar (Martin & Naples, 2008), the enamel reaches a maximum thickness of 20 and 30 microns respectively but closer to the base measures only four microns (Dumont et al., 2016). These ornithurines were much larger than any longipterygid, with H. regalis reaching an estimated twenty-one times the predicted mass of L. chaoyangensis (3,300 and 155 g respectively) (Bell & Chiappe, 2016; Dumont et al., 2016; Peters & Peters, 2009). Compared to extant taxa, L. chaoyangensis exhibits similar enamel thickness as insectivorous chiropteran molars (Table S1).

Geometric morphometric analysis

Five different dietary preferences were analyzed (Fig. 3) and Euclidian distances in multivariate space were measured between sampled longipterygids and other taxa (Table S5). The results reveal that the teeth of L. chaoyangensis are similar in morphology to the canine teeth of the carnivorous mammals Canis lupus, Genetta genetta, and Caracal caracal and the maxillary teeth of the carnivorous sauropsids Varanus cumingi, Heloderma suspectum, Tyrannosaurus rex, and Microraptor gui. Additionally, L. chaoyangensis shows similar morphology to the upper canines of the insectivorous chiropteran Diclidurus albus and maxillary teeth of the sauroposid Sphenodon punctatus. The results indicate the teeth of L. hani are similar to the maxillary teeth of the carnivorous sauropsids Varanus bengalensis and Caiman crocodilus, the upper canines of the carnivorous mammals Crocuta crocuta and Eira barbara, and the insectivorous chiropterans Hipposideros camerunensis, H. beatus, Mormoops megalophylla, and Cormura brevirostris (Figs. S3 and S4). The teeth in some of these taxa (e.g., Cai. crocodilus, Cro. crocuta, E. barbara) lack labiolingual compression and are basally expanded, similar to all extant taxa that plotted close to L. hani (Figs. S3 and S4).

Figure 3 Two-dimensional geometric morphometric analyses of the teeth of two longipterygids and other taxa.

(A) Using five different dietary preferences and grouping using convex hulls, the morphologically similar teeth are shown compared to Longipteryx chaoyangensis and Longirostravis hani. Interestingly, the tooth morphologies of animalivorous (e.g., insectivorous, piscivorous, carnivorous) taxa show much overlap. Factors such as the size of the predator, the ability or inability to orally process prey, and the strategy of prey acquisition can account for the differences and similarities between taxa. Both longipterygids are labeled in each analysis and represented by an asterisk. (B) Visualizations of the positive (max) and negative (min) PC1–3 values corresponding with both (PC1 × PC2 and PC1 × PC3) analyses with the mean morphology represented by blue and the minimum/maximum representations in red. Taxa sampled and protocols used can be found in the Supplemental Information.

In the PCA results, PC1 explains 44.01%, PC2 explains 25.36% and PC3 explains 13.75%, together representing most of the shape variation of the targeted tooth. PC1 describes tooth proportions, PC2 apicobasal curvature, and PC3 mesiodistal curvature. Deviations from the mean tooth morphology are captured by the PCAs (Fig. 3 and Fig. S3). Negative PC1 values represent a tooth that is relatively apicobasally elongate and mesiodistally narrow. Positive PC1 values represent a tooth that is apicobasally short and buccolingually wide. Negative PC2 values represent a tooth that exhibits slight apicodistal curvature, whereas positive PC2 values indicate greater apicodistal curvature. Negative PC3 values exhibit a morphology in which the distal margin is expanded midcrown. In contrast, positive PC3 values indicate concave mesial curvature at the midline (viewed laterally), giving a more exaggerated “curved tooth” morphology. Along all these three axes, the insectivorous and omnivorous taxa largely overlap with other dietary preferences, with the insectivores occupying a majority of the available morphospace.

Carnivores have negative PC1 values, indicating that their teeth are relatively more apicobasally elongate and mesiodistally narrow which might be related to enhanced slicing ability compared to teeth that are short and blunt (Fig. 3). Herbivores have negative PC2 values, indicating that they have less apicodistally curved teeth. Most piscivores have positive PC1 values and negative PC2 values with the exception of the yangochiropteran Noctilio leporinus, who unlike the other piscivorous taxa sampled, exhibits comparatively linear mesial and distal tooth margins (specifically in the canine), plotting within negative PC1 and positive PC2 morphospace. Dietary preferences are not obviously separated or distinguishable in PC3. In PC1, L. chaoyangensis is most closely aligned with carnivores and herbivores, in a region also overlapped by insectivores (Table S6). Its higher PC2 value slightly distinguishes it from the herbivores. Longirostravis hani is similarly contained within both the insectivorous and carnivorous morphospace along PC1 and PC2. Although unsurprising given the diversity of insect prey and modes of capture, the extensive morphospace occupied by insectivorous taxa makes direct associations difficult.

Results of the ordinary (i.e., non-phylogenetically corrected) Procrustes ANOVA indicate significant differences in tooth shape among carnivorous (F1,99 = 5.2627, P value < 0.01), omnivorous (F1,99 = 5.9402, P value < 0.01) and herbivorous (F1,99 = 5.3974, P value < 0.01) diets (Table S6). However, while considering the influence of phylogeny, the phylogenetic ANOVA results revealed that the differences in tooth shape are not significant in any of these diets (Table S6), ruling out strong predictive associations of tooth morphology in sampled taxa.

Rostral proportion comparisons

Longipterygid rostral proportions closely resemble those of the extant avian families Meropidae (e.g., bee-eaters), Mimidae (e.g., thrashers), and Alcedinidae (e.g., kingfishers) (Table S4). Plotting log skull length against proportional rostral length (Fig. 4) resulted in two clusters of longipterygids. Longirostravines plot strongly within the insectivorous cluster, whereas longipterygines are associated with insectivory, carnivory, and omnivory. Omnivory is observed throughout much of the available space. Insectivory, unlike omnivory, tends to diminish across sampled taxa once log10 skull length surpasses a value of 100 mm. The specialized dietary preference nectivory occupies a limited region with the group as a whole exhibiting the proportionately greatest rostra sampled but also failing to surpass a log skull length of 1.8 (~ 63 mm).

Figure 4 Skull proportions and dietary associations among extant birds.

Seven dietary regimes were tested against the relationship between log skull length and proportional rostral length length. Extinct taxa (longipterygids) are marked on the plot and form two loose but separate clusters between members of the Longirostravinae and Longipteryginae. Each species is represented by their primary diet, though this often does not encompass the full range of their diet. Within both figures, where longipterygids clump, adjacent arrows aid in differentiating them from other sampled taxa. For taxa included in this figure, see Table S4. Silhouettes of birds are for representation of general morphology at that point in the figure. Top left: Calypte costae, bottom left: Psaltriparus minimus, bottom right: Pelecanus occidentalis.

Phylogenetic generalized least squares regression (pGLS) indicates that both the relative rostral length and absolute skull length have significant associations with some dietary traits (Table S7). Rostral length (in models of the form “rostral length ~ skull length + dietary trait”) has a significant positive allometry with skull length, in which birds with longer skulls have proportionally longer rostra (coefficient = 1.24, P < 0.0001; Table S7). After accounting for this, residual rostral length has a significant relationship with invertivory (coefficient = 0.0325; P = 0.0041) and herbivory (coefficient = −0.0287, P < 0.0001; Table S7), indicating that invertivorous birds have proportionally longer rostra than expected for their skull length, whereas herbivorous birds have proportionally shorter rostra. Skull length on its own has a significant relationship with several diet categories: piscivory (coefficient = 0.1369; P < 0.0001) and insectivory (coefficient = −0.0588; P = 0.0037; Table S7), indicating that piscivorous birds are on average larger than non-piscivorous birds and that insectivorous birds are on average smaller than non-insectivorous birds. This suggests we can rule out a piscivorous diet for the smaller-bodied longipterygids (Fig. 4). Viewed together, skull length and proportional rostrum tend to show strongest ecological signal in sampled taxa for diets that contain invertebrates (e.g., foraging for insects, or aquatic mollusks, arthropods, or gastropods). Multivariate pGLS tests reveal rostral shape (i.e. laterally-viewed morphology) ~diet was significant for all dietary regimes except for carnivory (Table S7).

Discussion

Assessing the diet and subsequent ecology of extinct animals is challenging, especially when faced with combinations of features not observed in any living group. With Mesozoic birds it is tempting to suggest ecological roles based on comparisons with extant birds that are superficially similar, although with more detailed comparison these parallels may fall short, as is the case with the Longipterygidae which we attempt to address here. Most previous hypotheses concerning the diet and ecology of longipterygids made comparisons to extant birds based on their rostral elongation relative to other enantiornithines while ignoring the obvious limitation that these extant analogues lack dentition (or even pseudodentitions), the morphology of which is also a very distinctive feature of the clade.

Dentition restricted to the distal portion of the jaw in enantiornithines is not unique to longipterygids. Falcatakely forsterae from the Late Cretaceous of Madagascar has teeth restricted to the premaxilla (O’Connor et al., 2020). The holotype (and only known) specimen preserves a single conical tooth in the left premaxilla and the presence of additional teeth is uncertain (O’Connor et al., 2020). The rostral morphology in F. forsterae is disparate from longipterygids, being proportionately much deeper throughout its length.

In the Ornithuromorpha, a similar morphology also occurs in Mengciusornis dentatus (Wang et al., 2019). In M. dentatus, teeth are only present in the premaxilla and are absent from the dentary, and the rostrum is not significantly elongated suggesting that the rostrally restricted dentition in this taxon also evolved under selective pressures different from those driving longipterygid evolution. Notably the closest relative of M. dentatus, Schizooura lii, is edentulous suggesting the restricted dentition in this taxon represents a stage in the evolution of an edentulous rostrum in the Schizoouridae (Wang et al., 2019). Differences in appendicular proportions and pedal morphology (e.g., pedal phalanges decrease in length distally and metatarsal III extends beyond termination of metatarsals II and IV in M. dentatus) between M. dentatus and the Longipterygidae further indicate ecological differences between these two taxa.

This diversity necessitates a more holistic approach that utilizes comparative qualitative and quantitative analyses of extinct and extant taxa incorporating multiple aspects of their morphology to produce stronger predictions of the diet and possible ecology of the longipterygids. Below, we deconstruct previous predictions and synthesize new lines of evidence to infer the diet and possible ecology of the Longipteryginae and Longirostravinae. While no single morphological feature is capable of providing robust inferences with regards to diet, when used in combination, the morphology of the feeding apparatus and hindlimbs, the latter of which interacts directly with the substrate, can help to provide a rigorous hypothesis for the ecology of extinct toothed birds.

Examination of previous Longipteryginae predictions

Longipteryx chaoyangensis was originally described as a piscivore and was compared to extant piscivorous kingfishers (Coraciiformes: Alcedinidae) (Zhang et al., 2001) based on superficial similarities in rostral and appendicular proportions (Zhang et al., 2001). However, recent interpretations suggest longipterygids were insectivorous (Miller et al., 2022). Of the longipterygids, only L. chaoyangensis is known from multiple specimens; seven have been reported thus far including the holotypes of Camptodontus yangi (Li et al., 2010) and Shengjingornis yangi (Li et al., 2012). The former is demonstrated to be a junior synonym of L. chaoyangensis while the diagnosis of the latter fails to support the specimen as a distinct species and thus both are treated as junior synonyms of L. chaoyangensis herein (Stidham & O’Connor, 2021; Wang et al., 2014a, 2015; Yun, 2019).

To date, no direct evidence of diet in the form of in-situ digested remains have been reported in longipterygids or any other Jehol enantiornithine, despite the reports of stomach contents in most other avian lineages present in the Jehol Biota (O’Connor, 2019). In contrast, ingested fish remains are commonly recovered in the piscivorous ornithuromorph referred to the genus Yanornis, in the form of whole, undigested fish in the crop of some, a compacted pellet of fish bones in the crop of one specimen, and numerous specimens with macerated fish bones in the ventriculus (Wang et al., 2013; Zheng et al., 2014; Zhou, Clarke & Zhang, 2002; Zhou & Zhang, 2001). The absence of ingested fish remains in any known specimen of L. chaoyangensis as well as differences in dentition with other piscivorous birds have been used as an argument against the interpretation that L. chaoyangensis was piscivorous (O’Connor, 2019), although this hypothesis persists (Li et al., 2020; Zhou et al., 2021). The position of the toothrow greatly differs in longipterygids compared to other toothed piscivorous birds. Piscivorous bird taxa that have teeth, or pseudoteeth in the case of the pelagornithids (Louchart et al., 2018), exhibit them throughout most of the jaw with tooth loss only in the tip of the upper jaw in some (e.g., Y. martini), the opposite pattern observed in longipterygids in which tooth loss occurs in the caudal portion of the toothrow. In Y. martini the number of teeth is also increased relative to the plesiomorphic condition, with more teeth present than in any other Early Cretaceous bird (Zheng et al., 2014). Although the premaxilla is edentulous, the tooth count is similarly high in both the probable piscivorous genera Hesperornis and Ichthyornis, and the tooth row extends further caudally than in other toothed birds, nearly reaching the orbit (Field et al., 2018). Additionally, L. chaoyangensis exhibits tooth enamel twice as thick as known piscivorous Mesozoic birds (e.g., Hesperornis regalis, Ichthyornis dispar) (see results, Table S1). Altogether, the reduced dentition, distally-restricted location of the teeth in the jaw, and the unusually thick tooth enamel, do not support interpretations regarding a piscivorous diet for longipterygines (Stidham & O’Connor, 2021).

Interpretations regarding a plunge-diving feeding behavior (Zhang et al., 2001) are also unsupported upon closer examination. The presence of teeth distally restricted in the rostrum, as well as the angle of the rostrum itself, being slightly decurved, differ from the straight, edentulous bill morphology in piscivorous alcedinids required for their plunge-diving form of prey acquisition (Crandell et al., 2019; Crandell, Howe & Falkingham, 2019; del Hoyo, 2001; Zhang et al., 2001). Their straight bill morphology reduces drag when entering the water, whereas the decurved morphology of the dentary and the presence of large distally-restricted teeth in the jaw would both increase drag in L. chaoyangensis.

While the overall rostral morphology in longipterygines is dissimilar to plunge-diving piscivorous alcedinids, they do nonetheless exhibit some morphological traits similar to the Alcedinidae as a whole (e.g., elongate rostrum (Fig. 4), caudally deep shape of the bill). It should be noted that alcedinids are not restricted to riparian habitats (e.g., genera Halycon and Dacelo), but are also found within inland habitats where their bill morphologies, prey preferences, and hunting strategies differ from their predominantly plunge-diving piscivorous relatives (Brusaferro & Insom, 2009; del Hoyo, 2001; Winkler, Billerman & Lovette, 2020b).

Apart from previous hypotheses focused on piscivory, the observation of crenulations (fine serrations) on the distal margin of the premaxillary teeth in L. chaoyangensis led to the prediction that this taxon was hypercarnivorous relative to other enantiornithines and possibly fed on small vertebrates (O’Connor, 2019; Wang et al., 2015). Labiolingually compressed ‘blade-like’ teeth are commonly associated with carnivorous diets in both extant and extinct taxa (Holliday & Steppan, 2004). Though longipterygids are similar in size to some other generally animalivorous extant birds (e.g., Coraciiformes), the presence of unusually thick enamel is not associated with carnivory in paravians, being absent in both a microraptorine and A. huxleyi; taxa closely related to birds and documented to engage in animalivory (Li et al., 2020; O’Connor & Zhou, 2020).

Examination of previous Longirostravinae predictions

Hou et al. (2004) and Miller et al. (2022) compared L. hani to families within the extant order Charadriiformes, colloquially known as “shorebirds” (e.g., genera Haematopus and Tringa), that typically forage in littoral environments and wetlands utilizing a diverse set of bill morphologies depending on their diet (i.e., distally blunted bills used for prying, elongate tapering bills used for probing, or elongate rostrolaterally directed bills for sifting) (Lovette, 2016). This original interpretation was challenged by Morschhauser et al. (2009) and O’Connor (2019). Morschhauser et al. (2009) noted that while the elongate rostrum is somewhat reminiscent of some extant bird bills that engage in soft sediment probing, such taxa do so with the aid of a special form of cranial kinesis known as distal rhynchokinesis (Estrella & Masero, 2007; Zusi, 1984), which is predicted to be morphologically impossible in the larger longipterygines (O’Connor, 2019; O’Connor & Chiappe, 2011). While some form of limited distal flexion may have been possible in the more delicate rostrum of L. hani and other longirostravines (O’Connor & Chiappe, 2011), cranial kinesis is generally regarded as absent in enantiornithines and other basal birds (Hu et al., 2019; Wang et al., 2021, 2022). Furthermore, the Longipterygidae as a whole lack postcranial features associated with extant soft sediment probing birds, which also have proportionally long hindlimbs (Kilbourne et al., 2016), flattened, minimally-recurved pedal unguals (Manegold, 2006), elongated pedal digits (Lovette, 2016), and pedal phalanges that decrease in length distally (Hopson, 2001; Morschhauser et al., 2009; O’Connor, 2019).

Based on the arboreal morphology of the foot in R. pani, a bark-probing ecology was suggested (Morschhauser et al., 2009). However, detailed consideration also reveals incompatibilities with this prediction. Probing members within the extant families Certhiidae and Furnariidae tend to forage between and underneath bark as well as glean from leaf tufts and flower bases (Winkler, Billerman & Lovette, 2020c). In nearly all extant birds that occupy a niche primarily utilizing probing to obtain food, the bill tapers distally to a near point (Lovette, 2016). This tapered feature, like forceps, works well for precisely picking small insects from crevices. In contrast, the premaxilla and distal end of the dentary in the Longirostravinae (and all longipterygids) do not fully occlude, but instead partially angle away from one another, likely to account for the teeth. While the act of probing is physiologically plausible in the longirostravine subclade, we find it unlikely that it was the primary driver of the elongate rostral morphology for the reasons discussed above.

New observations on the Longipteryginae

Tooth morphology in L. chaoyangensis suggests an animalivorous diet. In the results of the GM analysis, tooth morphologies show associations with both carnivore and insectivore dietary regimes. In the closely associated extant taxa, particularly the carnivorous species (e.g., C. lupus, G. genetta, C. caracal, V. cumingi, H. suspectum), the distally-located teeth in the rostrum (e.g., canines in mammals, distally-located maxillary teeth in sauropsids) are apicodistally curved and mesiodistally narrow and used for piercing, slicing, and tearing. Longipteryx chaoyangensis also exhibits crenulations on the apical portions of the distal margins of the premaxillary teeth (Wang et al., 2015) formed by small denticles (Fig. 1C) (Wang et al., 2015). Similarly, carnivorous non-avian theropod dinosaurs (e.g., tyrannosaurids, dromaeosaurids) commonly have teeth that are crenulated (with denticles), either along a single margin or both (mesial and distal) (Torices et al., 2018). The prevalence of this feature in carnivorous dinosaurs suggests a correlation between their presence and certain carnivorous feeding strategies (Torices et al., 2018). Crenulations are notably absent in the known carnivorous and piscivorous A. huxleyi (Hu et al., 2009). The crenulations in L. chaoyangensis are likely functionally homologous to the denticles (crenulation vs denticle reflecting differences in scale, both referring to a serrated margin) in non-avian theropods (e.g., Dromaeosaurus albertensis). One study suggests serrations on the distal tooth margin optimizes tooth integrity in taxa that handle struggling prey (Torices et al., 2018). Additionally, Torices et al. (2018) demonstrate that the apical half of the distal margin of the teeth experienced the greatest amount of pressure and stress when biting into prey. This is the region of the tooth where the crenulations occur in L. chaoyangensis, suggesting this feature may have evolved in response to stress induced by struggling prey. Together, tooth morphology suggests insectivory and opportunistic carnivory for L. chaoyangensis. Comparisons with extant volant dentulous carnivorous taxa are limited, with few members of the yangochiropteran known to consume vertebrate prey (e.g., genera Noctilio, Vampyrum, Mimon, Phyllostomus, and Nycteris) (Vehrencamp, Stiles & Bradbury, 1977). Though plausible, it is difficult to determine what level of carnivory might have been present in this extinct taxon.

Thickened enamel in L. chaoyangensis alone gives weak support for a specific dietary regime, but when viewed together with tooth morphology, supports predictions for an insectivorous diet. Dental enamel, the hardest substance in the vertebrate body, promotes tooth rigidity when engaging in stressful force-producing behaviors, such as quickly biting down on tough materials, shaking prey, or biting to the point of occlusion (Dumont, 1995; Lucas et al., 2008; McCurry et al., 2015; Pampush et al., 2013; Rensberger, 1995, 1999; Strait, 1993; Ziscovici et al., 2014). Based on available data, L. chaoyangensis exhibits greater enamel thickness than most other Mesozoic birds and carnivorous non-avian theropods, with values similar to much larger carnivorous non-avian theropods (genera Allosaurus, Majungasaurus, Ceratosaurus), even though the predicted mass of L. chaoyangensis (~ 155 g) is 1.5 × 104 less than genera like A. fragilis (~ 1,814 kg) (Benson et al., 2014) (Table S1). The absence of relatively thickened enamel in closely related carnivores suggests that the thickened enamel in L. chaoyangensis did not evolve through pressures related to feeding on vertebrates.

Thick enamel is positively associated with greater fracture resistance (Sellers, Schmiegelow & Holliday, 2019). Among mammals, granivorous and frugivorous primates have caudally-located teeth (e.g., molars) with relatively thicker enamel to crush seeds and nuts (Olejniczak et al., 2008). Longipteryx chaoyangensis, lacking caudal teeth in the jaw, likely lacked the ability to orally process (i.e., chew or masticate food), a behavior considered absent in all birds. Enamel thickness variations in insectivorous chiropterans are associated with dietary preference (Barlow, Jones & Barratt, 1997; Dumont, 1995; Dumont et al., 2012). For example, chiropterans feeding on thicker chitinous exoskeletons (e.g., Coleoptera) tend to exhibit thicker tooth enamel than those feeding on softer-bodied insects (e.g., Neuroptera and Ephemeroptera) (Dumont, 1995; Evans & Wang, 2005; Strait, 1993).

Van Valkenburgh (1988) demonstrated that in animalivorous carnivorans, canines tend to break proportionally more often than incisors, premolars, or carnassials. These distally located teeth in the jaw break more often because they often make initial contact with prey items and are involved in puncturing, holding, and tearing actions. Thick enamel in the teeth of L. chaoyangensis may have compensated for similar stressors faced by extant predator canines, being in the same general position in the mouth (i.e., distal in jaw). These data further suggests that the teeth in L. chaoyangensis may have been adapted for behaviors such as piercing the thick carapace of hard-bodied insects.

Enamel thickness variations and dietary associations are also observed in extant sauropsids. Among carnivorous reptiles, proportionately thicker enamel is positively associated with proportionally larger prey items (i.e., often thicker integument) (Fitch, 1981; Sander, 1999; Rensberger, 1995, 1999). Invertivorous reptiles also tend to exhibit thicker enamel than carnivorous species (Sander, 1999). Experimental data in crocodilians (Sellers, Schmiegelow & Holliday, 2019) has also suggested that thicker enamel is associated with greater bite forces (Herbst et al., 2021). This is consistent with the prediction that L. chaoyangensis had a strong bite force relative to other similarly sized basal birds (Stidham & O’Connor, 2021).

Rostral proportions and body mass in L. chaoyangensis jointly suggest animalivory (Fig. 4). In most birds, the rostrum is the primary feature associated with feeding and prey capture (Pigot et al., 2020). Thus, the functional interpretation of the elongate rostra in longipterygines is that it was primarily an adaptation for specific feeding behaviors. However, it should be noted that bill morphologies are also shaped by selective pressures related to vocalization, thermoregulation, and feather maintenance (Bright et al., 2016, 2019; Navalón et al., 2019; Clayton et al., 2010; Clayton & Cotgreave, 1994). Among extant avian aerial insectivores, there are two predominant bill morphologies: those that are rostrocaudally short and mediolaterally wide (e.g., Eurylaimidae and Caprimulgidae), which facilitate funneling actions, directing prey items into the mouth (i.e., engulfing) (Beecher, 1962; Lederer, 1975), and those that are rostrocaudally elongate and distally tapered (e.g., Galbulidae, Momotidae, certain members of the Alcedinidae, Meropidae) which acquire prey items near the distal end of the bill and then move them back into the mouth (del Hoyo, 2002). Insectivorous birds with proportionally elongate rostra are better suited for capturing quick-moving prey. Elongate rostra can occlude their tips faster than mediolaterally-wide bills, facilitating the capture of faster-moving prey items (del Hoyo, 2002). These two hunting strategies are also inferred for insectivorous pterosaurs, exemplified by the mediolaterally wide-mouthed Anurognathidae and the elongate rostra of wukongopterids and darwinopterids, with different rostral morphologies hypothetically specialized to feed on different groups of flying insects (Bestwick et al., 2020; Clark & Hone, 2022). This same evolutionary pattern is also seen in extant carnivores such as canids and hyenids in that proportionally elongate gracile skulls are better suited for capturing faster moving small prey items (Freeman, 1984; Slater, Dumont & Van Valkenburgh, 2009).

Both Procrustes ANOVA and phylogenetic ANOVA results indicate significant differences between some dietary regimes when considering proportional rostral length with body mass. Dietary regimes of herbivory (P < 0.001) and invertivory (P = 0.004) signal stronger predictive power concerning associations between rostral length, skull length, and diet in our represented sampled taxon. Two specimens of L. chaoyangensis used in cranial analyses plot closely associated with dietary regimes of carnivory, omnivory, and insectivory within our analyses (Fig. 4). Our results, accompanied by comparisons between both proportional and absolute rostral length in longipterygids, suggest that the elongate rostrum in this group may have evolved in response to similar evolutionary pressures as galbulids, in-land alcedinids, and meropids. These groups feed on relatively fast-moving prey items like flying insects (e.g., odonates, neuropteridans, lepidopterans), as a primary, although not necessarily exclusive, food resource. When foraging, galbulids, alcedinids, and meropids catch prey items at the distal end of the bill, and then proceed to beat them against hard surfaces to fully dispatch them. This process rids insect prey of the wings and possible stingers and crushes the skull of vertebrate prey (del Hoyo, 2001, 2002; Fry, 1980). Perhaps thickened enamel reduced the chances of tooth damage when engaging in similar prey-dispatching methods in L. chaoyangensis. Acquiring prey items at the distal end of an elongate rostrum rather than closer to the face (as with a shorter rostrum) also protects sensitive body parts like the eyes, which are vulnerable to damage from struggling prey (e.g., interaction with wings, stingers, or claws) (Fry, 1980; del Hoyo et al., 1992, 2001, 2002). This strategy also allows time to determine prey palatability, where to strike next to incapacitate prey items, or to make-sure that hazardous features are destroyed (e.g., pincers, stingers) before prey is consumed (del Hoyo, 2002).

The dentary of L. chaoyangensis (the B. zhengi holotype specimen lacks a complete skull and is the only known specimen) exhibits a distally downturned morphology, not allowing rostral tip occlusion. The ventrally concave angle of the dentary in most members of this clade, unique among enantiornithines, is likely not a taphonomic artifact since it occurs in multiple specimens (O’Connor & Chiappe, 2011). This morphology may have evolved to compensate for the large, distally-restricted located teeth in the rostrum, thus allowing for near dental occlusion when the rostrum was held closed. In extant dentulous volant taxa (Chiroptera), precise occlusion of the teeth involved in prey-acquisition (e.g., distally-located incisors and canines in chiropterans) is crucially important for capturing fast-moving prey items, particularly in mid-flight due to the brief time frame in which prey can be caught and processed (Freeman, 1992).

In summary, these multiple lines of evidence support insectivory, and possibly animalivory, as the likely diet of L. chaoyangensis, as previously hypothesized (O’Connor, 2019; Miller et al., 2022). The tooth morphology, thick tooth enamel and rostral elongation exhibited in L. chaoyangensis may have been used in quick biting actions applied to prey items that could otherwise fracture less equipped teeth, such as thick chitinous carapaces or possibly to accommodate for stress-inducing intraspecific behaviors that incorporated the teeth. Possible prey items known from the Jehol Biota include coleopterans and odonates (Chang, Kirejtshuk & Ren, 2010; Lin, 1994; Zhang, 2000). Additionally, small vertebrates like juvenile birds, non-avian sauropsids, and amphibians may also have been consumed (Evans & Wang, 2010; Ji et al., 2002; Jin et al., 2006; Zhou & Wang, 2010; Zhou & Wang, 2017).

New observations on the Longirostravinae

The tooth morphology of longirostravines supports an animalivorous diet, specifically insectivory, with emphasis on puncturing, rather than cutting, actions. In the Longirostravinae, the teeth are peg-like in shape, lacking the apicodistal curvature observed in the Longipteryginae and potentially also the labiolingual compression, although small size and poor preservation makes the latter difference equivocal (Fig. 2) (Hou et al., 2004; O’Connor et al., 2009; O’Connor, Zhou & Xu, 2011). In the results of the GM analysis, tooth morphologies show associations with insectivory and carnivory. Closely associated extant taxa, particularly the carnivorous species (e.g., C. crocodilus, C. crocuta, E. barbara) exhibit maxillary teeth (sauropsids) or canines (mammals) with less labiolingual compression than taxa associated with more slicing actions (as those seen closer to morphospace to L. chaoyangensis). Like extant animalivorous taxa with similar tooth morphologies, we suggest the teeth of longirostravines were employed in puncturing and holding actions, likely targeting softer-bodied insect prey rather than employed for cutting or slicing actions required to feed on hard-bodied insects as inferred for the Longipteryginae. Apically blunt, conical canines in chiropterans require greater force to puncture the same substrates as apically sharper, serrated teeth (Freeman, 1998; Freeman & Weins, 1997). With estimated masses being less than half of those predicted for longipterygines, indicating inherently weaker bite forces, longirostravines likely had different dietary preferences and fed on smaller insects or those requiring less crack-propagating actions to penetrate (i.e., soft-bodied). Indeed, absolute size influences head size, bite force, and subsequent ecologies in small-bodied insectivores (Herrel et al., 2006).

Additional support for soft-bodied insectivory based on tooth morphology can be found in comparison with extant reptiles. In many reptiles, ontogenetic changes in diet are associated with changes in tooth morphology. Juveniles with apically blunter teeth (compared to mature forms) suited for gripping feed primarily on soft-bodied insects (e.g., genera Uromastyx, Lacerta), whereas adults mature into either heterodonts with larger labiolingually compressed teeth in the distal-most portion of the skull, facilitating the capture of thicker-shelled insects, or into complex-cusped herbivorous homodonts (Lafuma et al., 2021; Mateo et al., 1997; Sander, 1999). This ontogenetic change in tooth morphology and associated change in diet supports interpretations that proportionally small, unserrated, apically-blunt teeth in small taxa are utilized predominantly for soft-bodied prey acquisition.

Additionally, longirostravines clustered within insectivorous extant birds based on skull length and proportional rostrum length (Fig. 4). The absolute smaller skull length in longirostravines is suggestive of an insectivorous diet, as it is for extant taxa in the phylogenetic generalized least squares tests (Table S7). The sum of this evidence indicates insectivory as the primary diet regime for longirostravines with possible prey items in the Jehol Biota being softer-bodied groups like Lepidoptera, Neuroptera, and Ephemeroptera (Freeman & Lemen, 2007; Huang et al., 2007; Ren, Makarkin & Yang, 2010).

Implications of pedal morphology for foraging strategies

Pedal morphology helps further refine dietary and ecological predictions, as the pedes in extant birds are closely associated with locomotion, diet, development, and gripping ability (Backus et al., 2015; Botelho et al., 2014; Falk, Lamsdell & Gong, 2021; Höfling & Abourachid, 2021; Tsang & McDonald, 2018). As in other enantiornithines, the pedes in all longipterygids exhibit features suited to an arboreal lifestyle such as a reversed hallux (digit I) and large, curved unguals (Chiappe & Witmer, 2002; Dececchi & Larsson, 2011; Hou et al., 2004; Middleton, 2001, O’Connor et al., 2009; Zhang et al., 2001). Longipterygids have pedal phalanges that elongate distally such that the penultimate phalanges are the proximodistally longest within each digit, as observed in extant perching birds (Abourachid et al., 2017; Höfling & Abourachid, 2021; Hopson, 2001; Lovette, 2016; Morschhauser et al., 2009). Additionally, in all longipterygids the metatarsals are nearly equal lengths such that they terminate approximately at the same level, a feature that positively correlates to perching ability, with uniform termination facilitating an even foothold on sub-rounded substrates (e.g., branches) (O’Connor, Zhou & Xu, 2011; Wang et al., 2015; Zhang, 2006). These features indicate that longipterygids likely foraged within an arboreal environment.

Extant arboreal insectivorous birds with rostral elongation engage in perch-hunting behaviors such as sallying and hawking (e.g., Meropidae, Galbulidae) (Billerman et al., 2022), which may suggest longipterygids had similar prey-acquisition strategies. These foraging behaviors entail waiting on perches for prey items to pass by whereby the predator either flies in a straight line towards the prey, or drops down from the perch, flying more so in a loose “U” shaped trajectory. Occasionally, these extant groups capture flying insects with quick bill movements without vacating the perch (Pinheiro & Bagno, 2003).

The two largest longipterygids, the longipterygines L. chaoyangensisis and B. zhengi, both exhibit unique metatarsal proportions in that metatarsal IV exceeds metatarsal III in length (with the possible exception of the genus “Shenjingornis” PMOL-AB00179) (Li et al., 2012). This feature is not only absent in smaller longipterygids but is also not observed in other known enantiornithines (O’Connor, Zhou & Xu, 2011), which tend to exhibit proportionally larger pedes (Table 1). This relative extension of metatarsal IV also corresponds to an increase in the length of pedal digit IV, which is longer than digits II and III, a feature atypical for enantiornithines (Chiappe & Witmer, 2002). The unique, elongated, lateral pedal digit in the proportionately larger-bodied Longipteryginae may represent an adaptation for increased balance or grip, similar to other extant bird pedal morphologies in which the longest digit or digits with the greatest surface area are laterally located on the foot (e.g., zygodactyl morphology as in some members of Piciformes and Psittaciformes and syndactyl morphology as in some members of Coraciiformes, Piciformes) (Clark & O’Connor, 2021; del Hoyo, 2001, 2002; Winkler, Billerman & Lovette, 2020b, 2020d, 2020e). Increased lateral grip in the pedes of the larger longipterygines may support the prediction of differing diets from the smaller longirostravines, with larger members requiring greater grip and stability due to proportionally larger prey items being held in the jaw or different dispatching methods (e.g., shaking, hitting against substrates).

Table 1 Estimated body masses and pedal proportions in selected enantiornithines.

Scientific name	Specimen ID	Estimated mass (g)	Humerus length (mm)	Femur length (mm)	Tarsometatarsus length (mm)	Tibia/Tibiotarsus length (mm)	Pedal digit III length (mm) (Phalanxes 1–3)	Digit III: Tibiotarsus	Tmt: TbTa	
Longipteryx chaoyangensis	DNHM D2889	155.48	50.65	31	24	38	13.2	0.347368421	0.63157895	
Longipteryx chaoyangensis	IVPP V12325	119.34	43.48	28.77	19.2	30.7	Incomplete	Incomplete	0.62540717	
Longirostravis hani	IVPP V13309	42.6	24	19.43	13.74	25.17	Incomplete	Incomplete	0.54588796	
Rapaxavis pani	DNHM D2522	44.12	24.49	23	12.5	30	8.5	0.283333333	0.41666667	
Shanweiniao cooperorum	DNHM D1878	37.89	22.43	17.6	11.75	22.61	8	0.353825741	0.51968156	
Bohaiornis guoi	IVPP V17963	162.46	51.95	42.6	7.1	51.25	28	0.546341463	0.13853659	
Parvavis chuxiongensis	IVPP V18586	26.88	18.4	14.5	9.5	17.7	8.2	0.463276836	0.53672316	
Sulcavis geeorum	BMNH Ph-000805	134.57	46.6	41.3	24.85	47.3	21.8	0.460887949	0.52536998	
Parapengornis eurycaudatus	IVPP V18687	163.28	52.1	39.8	20.5	40.4	19.3	0.477722772	0.50742574	
Grabauornis lingyuanensis	IVPP V145595	75.94	33.5	31.3	19.4	36.2	16.8	0.464088398	0.5359116	
Junornis houi	BMNHC-PH 919a	48.62	25.9	23.4	17.5	28.1	13.2	0.46975089	0.6227758	
Sinornis santensis	BPV 538a	42.6	24	21	10	26.4	11.8	0.446969697	0.37878788	
Zhouornis hani	CNUVB-0903	155.21	50.6	43.5	25.8	51.2	21.9	0.427734375	0.50390625	
Protopteryx fengningnsis	PH1060-A	59.29	29.04	26.72	17.5	31.04	14.39	0.463595361	0.56378866	
Paraprotopteryx gracilis	STM V0001	38.39	22.6	22.2	15.5	26.3	14.9	0.566539924	0.58935361	
Huoshanornis huji	DMNM D2126	38.1	22.5	20.1	15.2	27.1	14.1	0.520295203	0.56088561	
Shenqiornis mengi	DMNH D2950/1	117.06	43	38.8	24.5	33	Incomplete	Incomplete	0.74242424	
Vescornis hebeiensis	CAS 130722	43.38	24.25	22.1	15.3	21.75	14.2	0.652873563	0.70344828	
Note:

Body mass estimates using equation given by Liu, Zhou & Zhang (2012). Among sampled enantiornithines, longipterygids have reduced pedal digit III-tibiotarsus proportions, with Longipteryx specifically exhibiting comparatively higher body mass as well. T, Tibiotarsus; TbTa, Tibiotarsus; Tmt, Tarsometatarsus. Pedal Digit III length entails the full length of phalanges 1–3.

Groups in which the surface area of the lateral pedal digit is maximized, either by elongating the outer digit or through fusion of the foot pads between digits III and IV, or even widening of the digit IV as in Fortipesavis prehendens (Clark & O’Connor, 2021), have improved grip and stability through increased surface area inducted friction. Such morphologies are common in extant birds that have proportionally reduced pedal and hindlimbs (e.g., families Ramphastidae, Alcedinidae, Meropidae, Galbulidae) (Backus et al., 2015; Höfling & Abourachid, 2021; Sustaita et al., 2013), as also observed in the Longipterygidae. Differences in foot morphology between extinct and extant birds with reduced pedes in proportion to body size may be due to developmental constraints in extant birds (Botelho, Smith-Paredes & Vargas, 2015; Clark & O’Connor, 2021).

Caveats of the predicted diets

Longipteryx chaoyangensis and L. hani are interpreted here as being primarily insectivorous, though carnivory is also considered plausible for the larger L. chaoyangensis. Predicted diets include hard parts that could readily become fossilized, as evidenced from direct evidence of animalivory in microraptorines and genera assigned to Yanornis from the same deposits (O’Connor, Zhou & Xu, 2011; O’Connor et al., 2019; Zheng et al., 2014) and evidence of ingested aquatic invertebrates preserved in Eoalulavis hoyasi from contemporaneous deposits in Spain (Sanz et al., 1996). Extant birds that feed on hard insects typically do so through the aid of ingested gizzard stones (Gionfriddo & Best, 1995). However, evidence of gizzard stones is altogether lacking in the Enantiornithes (O’Connor, 2019). This has been interpreted as indicative either of a primitive digestive system in this clade, inconsistent with the phylogenetic distribution of gastric mills in birds and dinosaurs as a whole, or a soft diet in enantiornithines (O’Connor & Zhou, 2020). Lack of in-situ vertebrate prey in longipterygines may be due to the limited or only opportunistic carnivorous behaviors, or alternatively, a lack of indicative fossil specimens.

Conclusions

Results suggest that members of the Longipterygidae exhibit dental features indicative of animalivory, with additional support for insectivory, especially in small-bodied species (Fig. 4). The larger Longipteryginae possess teeth that are apicodistally curved, labiolingually compressed, and crenulated similar to extant carnivorous taxa whose diets and ecologies reflect slicing actions and preferences for vertebrate prey items. However, the relatively thick enamel, inconsistent with other paravians that feed on vertebrates, and small body size suggest a diet of hard insects. In contrast, the Longirostravinae exhibit dental morphologies similar to many insectivorous chiropterans and generalist riparian animalivores that forage for both invertebrates and vertebrates. Longirostravinae teeth are apically rounded, conical (wider basally), and are well suited for puncturing with an emphasis on maintaining hold. Together with arboreal pedal morphologies and their overall smaller size, evidence suggests that the Longirostravinae primarily fed on soft-bodied flying invertebrates. The prediction of a primarily insectivorous diet for both subclades is accompanied by probable arboreal perch-hunting behaviors (Miller et al., 2022; O’Connor, 2019).

Our predictions are based on analysis of dental morphologies, additionally considering atypical enantiornithine pedal morphologies present in the clade, and contextually considering the rostral morphologies in extant birds previously used for morphological and ecological comparisons. Our analyses of these features provide support for the interpretation that longipterygids were primarily insectivorous. Animalivorous chiropterans do not possess proportionally long rostra (>60% of the skull) and all extant bird groups lack dentition, therefore, the dietary and ecological predictions of the Longipterygidae give us insight as to how morphologies from both groups (i.e., elongate bills in birds and piercing dentition in bats) may have functioned in tandem to fill a niche through a unique set of characteristics. In the future, the discovery of better-preserved feather impressions and possible stomach contents will aid interpretations regarding trophic niches, sexually dichotomous morphologies, and behaviors of this unique group of enantiornithines.

Supplemental Information

Supplemental Information 1 Examples of Tooth Landmarking in tpsdig264 of Sampled Taxa.

Three landmarks and two curves measured the morphological changes of the mesial and distal tooth margins in sampled taxa. From left to right, Artictus binturong, Caiman niger, Pteropus conspicillatus, Longipteryx chaoyangensis, “vectorized” Rhinolophus acuminatus. See Supplemental Information for landmarking protocols and taxon used.

Click here for additional data file.

Supplemental Information 2 Enamel Thickness Assessed Across Taxa.

With limited enamel thickness values available in the current literature, Longipteryx chaoyangensis plots near members of Chiroptera (mammals), and above some non-avian theropods and piscivorous avian theropods.

Click here for additional data file.

Supplemental Information 3 GMM Analysis with Labeled Taxa.

Using five different dietary preferences and grouping using convex hulls, the morphologically similar teeth are shown compared to Longipteryx chaoyangensis and Longirostravis hani. Interestingly, the tooth morphologies of animalivorous (e.g., insectivorous, piscivorous, carnivorous) taxa show much overlap. Factors such as the size of the predator, the ability or inability to orally process prey, and the strategy of prey acquisition can account for the differences and similarities between taxa. Both longipterygids are labeled in each analysis and represented by an asterisk.

Click here for additional data file.

Supplemental Information 4 Magnified View of Associated Taxa in the GM Analysis.

(A) Magnified section of PC1 × PC2 analysis with closely associated taxa compared to two sampled longipterygids. (B) Magnified section of PC1 × PC3 analysis with closely associated taxa compared to Longipteryx chaoyangensis. (C) Magnified section of PC1 × PC3 analysis with closely associated taxa compared to Longirostravis hani. (D) GM morphologies of sampled longipterygids and labeled taxa from A–C. As in Figure 3, mean tooth morphology is represented by blue and the specific taxon representation in red. All taxon sampled and protocols used can be found in the Supplemental Information.

Click here for additional data file.

Supplemental Information 5 Enamel thickness and known or predicted masses of selected taxa in the literature.

Click here for additional data file.

Supplemental Information 6 All sampled taxa included in GMM analysis, PGLS, and Procrustes ANOVA results.

Click here for additional data file.

Supplemental Information 7 Landmarking protocols for GMM analysis.

Click here for additional data file.

Supplemental Information 8 All sampled taxa included in bird skull and rostral analysis.

Click here for additional data file.

Supplemental Information 9 GMM Euclidian distances from sampled Longipterygids.

Click here for additional data file.

Supplemental Information 10 GMM analysis PGLS and Procrustes ANOVA results.

Click here for additional data file.

Supplemental Information 11 Bird skull pGLS and multivariate test results of skull length, rostral length, and shape.

Click here for additional data file.

Supplemental Information 12 Code.

All code used for GM, pgls, and phylogenetically-influenced analyses.

Click here for additional data file.

We would like to thank the editor and our reviewers for helpful suggestions that we believe strengthened our manuscript. Thanks to Dr. Matteo Fabbri for comments and advice. We would like to thank Stephen Rogers, Suzanna McLaren, John Wible, and Stevie Kennedy-Gold at the Carnegie Museum of Natural History for access to bird, mammal, and sauropsid specimens. We thank Dr. Emily Imhoff for access to the Cincinnati Museum Center specimens. We thank Ville Sinkkonen for his work as a paleoartist and his illustrations featured in Figures 1 and 2.

Additional Information and Declarations

Competing Interests

Author Contributions

Data Availability

The authors declare that they have no competing interests.

Alexander D. Clark conceived and designed the experiments, performed the experiments, analyzed the data, prepared figures and/or tables, authored or reviewed drafts of the article, and approved the final draft.

Han Hu performed the experiments, analyzed the data, authored or reviewed drafts of the article, and approved the final draft.

Roger B. J. Benson conceived and designed the experiments, performed the experiments, analyzed the data, authored or reviewed drafts of the article, and approved the final draft.

Jingmai K. O’Connor conceived and designed the experiments, authored or reviewed drafts of the article, and approved the final draft.

The following information was supplied regarding data availability:

The raw data and code are available in the Supplemental Files.

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
