# Peer review of "Reconstructing the dietary habits and trophic positions of the Longipterygidae (Aves: Enantiornithes) using neontological and comparative morphological methods"

_PeerJ, doi:10.7717/peerj.15139_

## Round 0.1 · original submission · Major Revisions

Dear authors,

Thank you for your submission to PeerJ. Based on comments from two reviewers and my own reading of the paper, I think that this paper will be suitable for publication in PeerJ following substantial major revisions. The main issues revolve around the paper needing more explanation of the methods used, likely additional analyses, and a restructuring of the results section (which is currently largely discussion). However, structuring the paper around a consistent outline as suggested by reviewer 1 would also help the paper tremendously. Both reviewers highlight these issues and please pay careful attention to their specific comments.

When you send in your revised version, please include an itemized response to reviewers document, a tracked changes version of your manuscript, as well as a clean version of your manuscript.

Please contact me if you have any questions.

Best,

Brandon P. Hedrick, Ph.D.

Major comments:

There is a major disconnect in the results section of the paper. It does not contain many results and many of the methods that you employ are not discussed. For example, the PCA and clustering from your GM analyses are not discussed at all in your results. The rostral proportions of birds and canine measurements are not discussed directly in the results, but instead are discussed more in the context of previous work. Much of the results section is not results at all, but instead falls heavily into discussion and likely outside of the scope of the paper. The sections ‘examination of previous Longipteryginae predictions’, ‘examination of previous Longirostravinae predictions’ and ‘review of Miller et al. (2022)’ must be moved to the discussion. They are not results and do not touch on any of the data collected in the paper. I think they might be shortened a fair bit as well since it does not all relate to diet, which is the primary goal of the paper. This causes it to distract from your main thesis. The results section I think must be heavily reorganized and the results of the analyses that you ran must be discussed.

This said, there are not many analyses mentioned in the methods. First, you do a PCA of the shape data. However, you need to do phylogenetic Procrustes ANOVAs of diet and shape, not just interpret the PCA. The ‘extant bird rostral proportions’ section of the methods does not include any analyses. It is not clear how these data were used. This is also true of the ‘canine measurements in extant chiroptera’ section. Statistical analyses should be used to here to compare across groups. Comparing measurements in the way that this seems to be done currently is qualitative and not as rigorous.

Insectivory is a form of animalivory. I would specify what you mean by animalivory. Similarly on line 78, you say ‘animalivorous or hypercarnivorous’, but these terms are not mutually exclusive.

The ‘Review of Miller et al. (2022)’ section seems like maybe it would be a better fit for a response to their paper than embedded in a GM paper at PeerJ.

Line by Line:

Line 56: This sentence is a bit confusing. The two terms are ‘inferred ecology’ and ‘predicted diet’? Maybe reword just to clarify.

Line 69–71: Should this be a separate paragraph?

Line 88: R. pani is from 2009, so not too newly discovered.

Line 111: GMM is usually for ‘geometric morphometric methods’ with GM being ‘geometric morphometrics’

Line 117–119: What does this mean? You weren’t able to see them in the photo? This needs to be explained more clearly.

Line 117: tpsDIG2 is fine. The 64 is just 64 bit. Capitalize DIG

Line 125: include version numbers for R packages

Line 136: That’s higher than the actual accuracy of digital calipers. I’d just state to 0.1.

Line 153: Do you mean ‘pteropodids’?

Line 177: You use both ‘apicobasally’ and ‘apicalbasally’ in this line of text. Probably apicobasal is better, but so long as it is consistent, either is fine.

Line 187: ‘positive PC1 values’

Line 340–343: This is an interesting point, although birds have a much wider dietary range than most bat families (the exceptions being pteropodids and phyllostomids). So you’re really only looking at differences in insectivory when you’re comparing with bats.

Line 349: ‘generalists’

Line 350: Probably ‘insectivory’?

Line 437: This is really the beginning of the results. The stuff up until this point was a combination of introduction and discussion.

Line 440: How does beak shape relate to keratin versus bone in this case?

Line 438–469: What part of this is new observations? This also reads like discussion. You should describe the overarching trends in your pca and your clustering here. And as I suggest, this is where your ANOVA results would belong.

Line 628: ‘sagittally’

Line 641–642: Although many bat species use their uropatagium to capture prey and then transfer the prey item to their mouth

Line 724: ‘in certain aspects’

Line 736–747: I feel like this paragraph is too speculative. I would advocate removing it.

Line 756: Grammar

Line 761–763: I think this is a bit speculative too, but could be supported with references.

Reviewer 1 ·

Basic reporting

1. Basic Reporting:
The English language is for the most part clear, but I have highlighted misspellings and some other problematic phrasing issues in detail below.
The introduction is adequate, but I believe that it should be expanded and refocused a bit with information currently provided and discussed later in the manuscript (e.g., the Miller et al. study and its results being included in the introduction).
The figures are relevant. See my comments below about suggested color changes and alterations to data analyses to provide stronger results to the reader.
The raw data are supplied, but as I note below, the authors need to state how they achieved 0.01 mm accuracy with digital calipers (that typically have a +/- 0.03 mm accuracy). Their data perhaps should be rounded to 0.1 mm, or have that error propagated in their analyses.

Experimental design

2.Experimental design
Yes, the study is within the scope of the journal.
The research question is well defined.
As noted below, I have questions about some of the data acquisition (caliper accuracy, where measurements were taken on the tooth) and the phylogenetic linkage of some of the presented data. Those issues need to be addressed in revision.

Validity of the findings

3. Validity of Findings
The underlying data have been presented along with the results of various analyses. As I have noted, I believe that further examination of the data (with respect to body mass or phylogeny) could provide invaluable information to test the hypotheses that the authors are examining. Does enamel thickness scale with body mass among the taxa sampled? Do the ecological groupings still exist after taking into account phylogenetic relatedness? Those questions need to be addressed before drawing conclusions from some of the graphs and charts presented.

I also think a second manuscript possibly could be pulled out of this one focusing on the bat canine morphology and its link to ecology/diet in bats. That publication would help to support the statements and correlations focused on here.

Additional comments

4. General comments
The authors have taken on an interesting topic in reconstructing the dietary preferences of an extinct bird group, and they have taken a unique approach to comparing the teeth to those of extant volant bats. However, I think this study is a bit incomplete.
In particular, the structure of the paper (with its repeating topics) needs revision and restructuring, extraneous sections (like on the tail feathers) can be cut or significantly reduced in length, and the data presentation (data visualization) can be improved. I do not believe that their conclusions are fully supported currently by their data and analyses, much like the Miller et al. paper that they extensively criticize.
The authors need to explain why bat canines, in particular, are a good model for extinct bird teeth, and why they warrant study. Are bat teeth relevant to understanding bird tooth function? That foundation is not established in the text. (see comment above)

The authors need a simple outline of their project and need to follow it. For example, talk about the foot and that it indicates the birds are arboreal, then move to another topic. Don’t keep coming back to the same topic repeating the same data and hypothesis.
Proposed outline: State the question you are researching. Talk about how others have addressed this before (condense your attack on the Miller paper here, tell us how you are remedying the flaws of that study). Talk about how you are approaching the problem and why you are choosing bats to do it with. Present your methods and new data. Show the results of your new analyses. Discuss your new data and how it relates to supporting or refuting your and previous hypotheses (that are brought up in the introduction). End.

Several areas of hypothesis exploration have not been addressed by the authors. They have limited the scope of their comparisons and seem to have concluded without conclusive examination that a form of carnivory is the diet. Their conclusion appears as tenuous as the ones they attack by Miller et al. Herbivory (frugivory/granivory) is not explicitly excluded in their study, and would seem a probable hypothesis given the forested setting of the birds. Only the length of the rostrum and two-dimensional shape of the teeth are examined here. I do not know if a hyoid apparatus is preserved in these taxa, but that could provide insight into feeding. If present, the apparatus should be discussed (do Miller et al. discuss it?). The authors also do not state if a retroarticular process is present in these taxa, nor its length (do Miller et al discuss this?). Some enantiornithines have that process, and it has a range of functions among living birds, including gaping (opening the mouth in a substrate) and filterfeeding (some ducks have a long rostrum too). Are there any other aspects of skull morphology preserved that could indicate skull function, bite force, or function? The authors mention the diverse Jehol flora, but plant products are never examined as potential dietary items here despite being known dietary items in enantiornithine outgroups Jeholornis and Sapeornis. There were plenty of seed-bearing cones, small fruits, and other plant parts available for these birds (and the diverse insect fauna), and presumably spread across a degree of hardnesses and nutritional qualities. Many extant arboreal long rostrumed birds consume fruits/seeds and insects. The authors focus on just the “prey capture” hypothesis and not potential plant part consumption (which could also explain the absence of preserved dietary items, if for example, soft seeds from hard cones were ingested). Insects were exploiting the cones of the dominant trees in that environment. Why not the birds? Perhaps consulting with a paleobotanist working on the Jehol Biota might provide insight.

Looking at some of the graphs, it appears that there is almost certainly a phylogenetic component to the data distributions (Figure 5 and 6). Drawing ecological conclusions from such data are highly suspect and the authors should remove/account for the phylogenetic component to see if the ecological groupings remain. If they do, then hypotheses about the diet of these extinct birds can be made. While the authors frequently plot absolute sizes for comparisons, I think comparing function of smaller and larger teeth solely from their shape and not their size can lead to erroneous conclusions. A 4 mm tooth has a different end result and abilities than an identically shaped 5 cm one in terms of obtaining dietary items. Scale can matter in function.

Did you sample only the upper canine of the bats, or the upper and lower? Why only one if you chose only one?
Could the rostrally restricted dentition function similar to a hooked beak present among many extant bird clades? Otherwise, there is no hooked beak morphology known among enantiornithines (correct?).
Interpreting what the thick enamel in one extinct bird species means is difficult. Mammals with their prismatic enamel and Hunter-schrager bands are constructed very different from the teeth of reptiles and have different mechanical properties. Their structural strength for a given thickness should not be directly comparable between the groups (mammals stronger). Using mammals as analogies for birds will have its limits. So using extant bats is not necessarily going to give you an answer applicable to birds. You need to scale your enamel thickness data to body mass to see if there is a potential link. Explain to the reader why this comparison is valid and useful. Why are you not looking at the incisors?

Specific comments:

There are many examples of awkward word choices or phrasing throughout the manuscript: ex. Line 39, “recent paleontology” What does that refer to? Recently published papers? You cite a nearly 50 year old paper.
Ex. Line 42/45, “predicting” We do not predict the diet of an extinct organism. We can try to reconstruct it or hypothesize what it was.

Line 44, what are the differences between burrowing, fosssorial, and subterranean. Those would seem to be largely identical.

Line 73, change the taxonomy. The traditional Coraciiformes are an order, but also the traditional grouping Coraciiformes is not monophyletic.

Line 110/111. List the city and country where the Carnegie is located.

Line 114/115. Alligator is not a squamate. You have 114 specimens which includes 2 alligators and a doubling of a varanid. If you have 112 species of squamates and mammals, how do you get 114 specimens total? Should it be 116?

Line 115, write out non-measurement numbers under 10. “Eight”

Line 117, are those programs? They should have citations for their publication.

Line 123, Change to “All of the”

Line 128/129 Citation for that database?

Line 135/136. State the city/country for the additional museum collections.

Line 136. What calipers were used? Most commercially available calipers are only accurate to +/- 0.03mm. That is why most workers round to the nearest 0.1mm (because of the built in error range of 0.06 mm). Did you propagate that measurement error in your analyses? Does rounding to 0.1 mm impact your analyses?

Line 148, What direction is the proximal root margin? Do you mean mesial surface?

Line 149/150. Canine height? So you measured the crown height plus some of the root? What is the cranial junction? It is unclear what you measured and if that is a consistent measurement across taxa. Why measure more than the crown? Is that consistently measured to homologous points across all specimens?

Line 151, what is the midline of a tooth? In what direction? Are you talking about the diameter of the crown? Unclear.

Section starting line144. Why were only the canines examined? Wouldn’t the incisors be better set of functional comparisons for the rostrally restricted teeth in the birds? Canines are a single tooth per bony element unlike the bird teeth which are repeating similar structures. Please explain why canines were chosen for comparison and not incisors or a combination of those teeth?

Line 199, correct to “Alcedinidae”

Line 205, You do not consider the fish bones with Piscivorenantiornis to represent diet?

Line 232, Do you mean curved? It isn’t recurved.

Line 237, What is the rostrum length percentage in kingfishers? Is it a similar 60%? You say the proportions are the same, but provide no data or references.

Paragraphs starting line 242, What is the relevance of the pygostyle to dietary choice? You can cut this section.

Line 269, not ‘evolutionary pressure’ It is natural selection.

Line 312/313. Are you talking about teeth? Then say teeth.

Line 337, distally? Rostrally? Caudally?

Line 380, Zygodactyl birds are able to change their pedal digit IV orientation from dorsal to plantar and back. It isn’t frozen in one position.

Line 381, toes face dorsally/distally, or plantarly, not craniocaudally.

Line 387, owls swallow small prey whole. They tear apart larger prey. You cannot say they only eat the head or entire body as a single unit. Eagle owls do not swallow the foxes they kill whole.

Line 391, what is pedes?

Lie 420/421. Provide a reference or data for that statement.

Line 468, what evidence is there for rapid closure? Is this just speculation? (see comment above about discussing other aspects of skull function)

Line 476, what are the similarities? Detail that. Add it to the introduction as to explain why you are doing these comparisons in the first place.

Line 507 del Hoyo et al., and you should probably cite the current online version of the book series.

Line 508, If you mean ‘serration’ among mergansers, then say that. Most anatids do not have serrations on their bills.

Line 530 paragraph, What is the relevance of this to diet? You need to tie in these random paragraphs and sets of data into a coherent structure.

Line 554, The shapes may be similar, but the sizes are not. The bird teeth are tiny as compared to the mammals and dinosaurs. You cannot assume function is the same across all scales of size. If so, you need to show that small teeth function identically to much larger teeth of the same morphology. Those tiny teeth might barely be able to break the surface of my skin, but a wolf certainly can because of the bite force (size) behind it.

Line 585, Many birds do orally process food. Finches, parrots and other birds with large tongues, use their cranial kinesis and bill modifications to peel seeds or fruits, and sort out debris from their diet.

Line 589/590, why would hard plant parts (cones, seeds) not result in the same selection on tooth enamel? Why only vertebrates and insects?

Line 604 paragraph and the following paragraph, This can be cut. It is not relevant to diet.

Line 662/664, is there a correlation? Can you predict potential prey size based on body mass estimates for the birds? If so, then do that.

Line 668, what is a ondontid?

Line 677, what direction is distal within the skull? Rostral/caudal/dorsal/ventral/medial/lateral.

Line 684, What genus is V? Write it out here.

Line 738, 31 genera of plants.

Figure 2. the coauthor gave permission to use a photo. Is it published elsewhere?

Figure 3, I recommend changing the colors. The current red/green spectrum will not be visible to most color blind readers. In addition, it is hard to discern even for me in its current state. Maybe bold the edges of the individual morphospaces. Maybe plot panel B on the graphs in panel A for better visualization as well too.

Figure 5, clearly shows that there is a strong phylogenetic component to rostrum size and proportion. You need to remove the phylogenetic aspect to derive any conclusions from these data. Also you should remove absolute size as a criterion in the graphs as well. (but you are making my point from above that size does matter in these interpretations of potential diet)

Figure 6. You need to remove size again for any meaningful comparisons.

Figure 7. This needs to be scaled against overall body mass of the individuals for any pattern to be identified.

Reviewer 2 ·

Basic reporting

This study investigates the possible diet and ecology of the Longipterygidae, a highly distinctive yet puzzling group of Mesozoic birds. The authors are to be commended for their holistic perspective on the topic, and I find their methods and conclusions to be generally reasonable. I do have some comments about certain specifics of the reporting and interpretations presented in this manuscript, as well as a few suggestions for additional literature to cite.

Throughout the manuscript, animalivory is sometimes used as a broad term covering insectivory, piscivory, and carnivory (e.g., Line 479), as is consistent with established terminology. Other times, however, it seems to be used as a distinct category from insectivory and other specific forms of animalivory (e.g., Lines 384 and 443, Fig. 5). In the latter examples, I take it that “animalivory” is intended to cover eating both invertebrates and vertebrates, in contrast to insectivores that specialize in invertebrate prey. Although understandable, I suggest that this usage should be explained in the beginning of the manuscript and kept consistent. Alternatively, or in addition to this, perhaps “general animalivory” could be used for the latter concept.

The sections reviewing and responding to previous studies are currently inserted in the middle of the results section, creating an abrupt break in the reporting of the authors’ own findings and observations. I recommend moving these sections to the end of the results, or even to the discussion section. The section on pseudodental structures in extant birds also reads more like discussion material.

Related to that section, further examples of crown-group birds with pseudodental structures that may be worth citing include African barbets of the genera Tricholaema and Lybius (Short and Horne, 2002), plantcutters (Phytotoma; Snow, 2004), the Tooth-billed Bowerbird (Scenopoeetes dentirostris; Frith and Frith, 2009), and the extinct moa-nalo (Thambetochenini, Anatidae; Olson and James, 1991). Korzun et al. (2004) is a notable reference for the function of the bill crenulations in motmots.

Orkney et al. (2021) might be an important reference to cite regarding the importance of holistic approaches to studying avian ecomorphology.

Line 91: Wang et al. (2013) also briefly disputed mud-probing habits in longirostravines on the basis of pedal morphology.

Lines 202–203: Is it accurate to say that Shengjingornis is “typically regarded” as a synonym of Longipteryx? The only previous study that I recall doing so is the cited Stidham and O’Connor (2021), which did not elaborate further on this decision. In contrast, the recent Miller et al. (2022) and Wang et al. (2022) both retained Shengjingornis as a distinct genus (and the phylogenetic analyses of the latter found it to be only distantly related to longipterygids).

Wang et al. (2014) and Wang et al. (2015) would also be appropriate to cite here regarding the assignment of “Camptodontus yangi” to Longipteryx.

Lines 265: Liu et al. (2012), which was mostly concerned with estimating body masses of fossil birds, is an odd reference to cite for the absence of rectricial bulbs in enantiornithines. Maybe Clarke et al. (2006) and Wang and O’Connor (2017)?

Line 416: As further support, Bright et al. (2016) and Bright et al. (2019) found similar complex relationships between bill morphology and diet.

Lines 757–759: I suggest citing a reference or two for these previous interpretations of enantiornithine digestive systems. (O’Connor and Zhou, 2020 seems appropriate.)

Literature cited:
Bright, J.A., J. Marugán-Lobón, S.N. Cobb, and E.J. Rayfield. 2016. The shapes of bird beaks are highly controlled by nondietary factors. PNAS 113: 5352–5357. doi: 10.1073/pnas.1602683113

Bright, J.A., J. Marugán-Lobón, E.J. Rayfield, and S.N. Cobb. 2019. The multifactorial nature of beak and skull shape evolution in parrots and cockatoos (Psittaciformes). BMC Evolutionary Biology 19: 104. doi: 10.1186/s12862-019-1432-1

Clarke, J.A., Z. Zhou, and F. Zhang. 2006. Insight into the evolution of avian flight from a new clade of Early Cretaceous ornithurines from China and the morphology of Yixianornis grabaui. Journal of Anatomy 208: 287–308. doi: 10.1111/j.1469-7580.2006.00534.x

Frith, C.B. and D.W. Frith. 2009. Family Ptilonorhynchidae (bowerbirds). Pp. 350–403, in J. del Hoyo, A. Elliott, and J. Sargatal (eds.), Handbook of the Birds of the World. Volume 14. Lynx Edicions, Barcelona.

Korzun, L.P., C. Érard, and J.-P. Gasc. 2004. Morphofunctional study of the bill and hyoid apparatus of Momotus momota (Aves, Coraciiformes, Momotidae): implications for omnivorous feeding adaptation in motmots. Comptes Rendus Biologies 327: 319–333. doi: 10.1016/j.crvi.2004.01.006

Miller, C.V., M. Pittman, X. Wang, X. Zheng, and J.A. Bright. 2022. Diet of Mesozoic toothed birds (Longipterygidae) inferred from quantitative analysis of extant avian diet proxies. BMC Biology 20: 101. doi: 10.1186/s12915-022-01294-3

O’Connor, J.K. and Z. Zhou. 2020. The evolution of the modern avian digestive system: insights from paravian fossils from the Yanliao and Jehol biotas. Palaeontology 63: 13–27. doi: 10.1111/pala.12453

Olson, S.L. and H.F. James. 1991. Descriptions of thirty-two new species of birds from the Hawaiian Islands: Part 1. Non-Passeriformes. Ornithological Monographs 45: 1–88.

Orkney, A., A. Bjarnason, B.C. Tronrud, and R.B.J. Benson. 2021. Patterns of skeletal integration in birds reveal that adaptation of element shapes enables coordinated evolution between anatomical modules. Nature Ecology and Evolution 5: 1250–1258. doi: 10.1038/s41559-021-01509-w

Short, L.L. and J.F.M. Horne. 2002. Family Capitonidae (barbets). Pp. 140–219, in J. del Hoyo, A. Elliott, and J. Sargatal (eds.), Handbook of the Birds of the World. Volume 7. Lynx Edicions, Barcelona.

Snow, D.W. 2004. Family Cotingidae (cotingas). Pp. 32–109, in J. del Hoyo, A. Elliott, and J. Sargatal (eds.), Handbook of the Birds of the World. Volume 9. Lynx Edicions, Barcelona.

Wang, M., Z.-H. Zhou, J.K. O’Connor, and N.V. Zelenkov. 2014. A new diverse enantiornithine family (Bohaiornithidae fam. nov.) from the Lower Cretaceous of China with information from two new species. Vertebrata Palasiatica 52: 31–76.

Wang, W. and J.K. O'Connor. 2017. Morphological coevolution of the pygostyle and tail feathers in Early Cretaceous birds. Vertebrata PalAsiatica 55: 289–314. doi: 10.19615/j.cnki.1000-3118.170118

Wang, X., L.M. Chiappe, F. Teng, and Q. Ji. 2013. Xinghaiornis lini (Aves: Ornithothoraces) from the Early Cretaceous of Liaoning: an example of evolutionary mosaic in early birds. Acta Geologica Sinica 87: 686–689. doi: 10.1111/1755-6724.12080

Wang, X., C. Shen, S. Liu, C. Gao, X. Cheng, and F. Zhang. 2015. New material of Longipteryx (Aves: Enantiornithes) from the Lower Cretaceous Yixian Formation of China with the first recognized avian tooth crenulations. Zootaxa 3941: 565–578. doi: 10.11646/zootaxa.3941.4.5

Wang, X., A. Cau, X. Luo, M. Kundrát, W. Wu, S. Ju, Z. Guo, Y. Liu, and Q. Ji. 2022. A new bohaiornithid-like bird from the Lower Cretaceous of China fills a gap in enantiornithine disparity. Journal of Paleontology advance online publication. doi: 10.1017/jpa.2022.12

Experimental design

As acknowledged by the authors (Lines 162–168), enamel samples taken in the previous studies consulted were not necessarily consistent with one another in terms of anatomical location. Additional discussion on how these discrepancies might influence the results and limit interpretations in the present study would benefit this manuscript.

Validity of the findings

Lines 178–179 and 186: The descriptions of low vs. high PC2 values may have been flipped here; in Fig. 3B, high PC2 is shown corresponding to stronger tooth curvature.

Line 179: Looks like “apicodistal” instead of “apicomesial” was intended here.

Line 180: Similarly, I would consider the curvature described here to be “distally” instead of “mesially” oriented. Since mesial means “towards the mandibular symphysis”, mesially-oriented curvature brings to mind forward-curving teeth.

Line 256: Shanweiniao is said to be the only longipterygid that preserves rectrices, but in the original description of Longirostravis, Hou et al. (2004) reported, “A pair of central rectrices, similar to those of the enantiornithine Protopteryx, appear to be present.” Has this been disputed?

Line 271: It may be true that plunge-diving coraciiforms lack ornamental tail feathers, but this might not be generalizable to all extant birds. Glancing over the cited reference Felice and O’Connor (2014), I cannot find where this claim is made. A notable example to the contrary is seen in tropicbirds (Phaethontidae), which have very elaborate tail plumage and hunt primarily by plunge diving (Orta, 1992).

Line 462: It is debatable whether dimorphodontian pterosaurs were insectivorous; evidence from tooth wear, for example, suggests Dimorphodon fed primarily on vertebrates (Bestwick et al., 2020). Maybe wukongopterids (darwinopterids) would be a more suitable example of long-snouted, insectivorous (or at least mixed insectivorous/carnivorous) pterosaurs (Lü et al., 2011; Bestwick et al., 2020)?

Line 496: I suggest selecting another example for enamel thickness variation in chiropterans, because Cebus is a primate instead of a chiropteran.

Line 525: Schuchmann (1995) described male tooth-billed hummingbirds hunting crevice-dwelling spiders, not flying insects. This is also mentioned in the Schuchmann and Boesman (2020) Birds of the World profile for the species. I would therefore not consider this support for the presence of tooth-like rostral projections as an adaptation for capturing flying insects.

Expanding on the authors’ point on p. 16 regarding the effects of taxon omission in the pedal analysis of Miller et al. (2022), it is striking that only one of the extant taxa sampled for the “perching” category in that analysis is anisodactyl (Opisthocomus). The others are zygodactyl (Cuculidae, Psittaciformes) or semi-zygodactyl (Musophagidae). All other anisodactyl birds in the dataset were considered terrestrial, raptorial, or scavenging, which would have conceivably influenced the results and interpretations of that study.

Literature cited:
Bestwick, J., D.M. Unwin, R.J. Butler, and M.A. Purnell. 2020. Dietary diversity and evolution of the earliest flying vertebrates revealed by dental microwear texture analysis. Nature Communications 11: 5293. doi: 10.1038/s41467-020-19022-2

Hou, L., L.M. Chiappe, F. Zhang, and C.-M. Chuong. 2004. New Early Cretaceous fossil from China documents a novel trophic specialization for Mesozoic birds. Naturwissenschaften 91: 22–25. doi: 10.1007/s00114-003-0489-1

Lü. J., L. Xu, H. Chang, and X. Zhang. 2011. A new darwinopterid pterosaur from the Middle Jurassic of western Liaoning, northeastern China and its ecological implications. Acta Geologica Sinica 85: 507–514. doi: 10.1111/j.1755-6724.2011.00444.x

Orta, J. 1992. Family Phaethontidae (tropicbirds). Pp. 280–289, in J. del Hoyo, A. Elliott, and J. Sargatal (eds.), Handbook of the Birds of the World. Volume 1. Lynx Edicions, Barcelona.

Additional comments

Line 40: Should be “Hesperornithidae” instead of “Hesperonithidae”.

Line 73: Should be “order Coraciiformes” or “family Alcedinidae” instead of “family Coraciiformes”.

Line 113: I think either “Reptilia” or “Sauropsida” was intended here instead of “Squamata”. Squamata is usually ranked as an order, so saying that 3 orders of squamates were sampled sounds strange, but it would make sense for Reptilia/Sauropsida (covering the orders Rhynchocephalia, Squamata, and Crocodilia).

Line 116: Should be “a non-longipterygid enantiornithine” instead of “non-longipterygid enantiornithines”, as according to the supplementary table, only one (Sulcavis) was sampled.

Line 122: Should be “Hendrickx et al. (2015)” instead of “Hendrickx et al. (2016)”.

Line 160–161: Based on the taxon sampling, specifying “non-avian theropods” and “non-archosaurian reptiles” instead of just “theropods” and “reptiles” would be appropriate for avoiding redundancy with the other listed taxa.

Line 164: It is stated that “apes” were excluded from the study, but the example given (Papio cynocephalus) is not an ape. Maybe “primate” or “anthropoid” was intended?

Line 202: Should be “Shengjingornis” instead of “Shenjingornis”.

Line 212: Should be “hesperornithiform” instead of “hespeornithiform”.

Line 222: Hesperornis and Ichthyornis were not both 22 times the body mass of Longipteryx, so I suggest rewording this sentence. Additionally, the listing of 155 g and 3300 g presumably refers to the estimated body masses of Longipteryx and Hesperornis respectively, but the current wording may lead readers to think that one of those is a mass estimate for Ichthyornis.

Line 223: Should be “Bell and Chiappe, 2016” instead of “Bell and Chiappe, 2015”.

Line 284: Should be “O’Connor and Zhou, 2020” instead of “O’Connor and Zhou 2019” for the final, paginated version. (This needs to be changed in the reference list, too.)

Line 339: Should be “significant importance” (or just “importance”) instead of “significance importance”.

Line 389: Should be “longipterygids” instead of “longipterigds”.

Line 424: Should be “resemble” instead of “resembles”.

Line 428: Should be “rectricial feathers” (or just “rectrices”) instead of “rectrices feathers”.

Line 463: Should be “Bennett” instead of “Bennet”.

Lines 555–558: The categorization of listed taxa here is a bit odd. It seems unnecessary to list “hypercarnivores” separately from other “carnivores”, or to mention “crocodilians” as their own category. I think “extant carnivores and insectivores” (with select examples for each) would be sufficient.

Line 556: Martes pennanti has been removed from the genus Martes and is now Pekania pennanti (note also the spelling of the species name) (Koepfli et al., 2008; Sato et al., 2012).

Lines 595 and 717: Should be “phalanges” instead of “phalanxes”.

Lines 597, 614, and 625: Should be “Höfling and Abourachid, 2021” instead of “Höfling and Abourachid, 2020” for the final, paginated version.

Line 608: Is syndactyly present in Psittaciformes? They don’t seem to be included among syndactyl taxa in the reviews by Botelho et al. (2015), Clark and O’Connor (2021), and Höfling and Abourachid (2021).

Line 611: Should be “widening of digit IV [four]” instead of “widening of the digit three”.

Lines 631–633: Is this meant to say that longipterygines are similar to other longipterygids in lacking distal occlusion of the rostrum, despite having greater absolute rostral length? Otherwise, it is not immediately clear how greater rostral length contrasts with the lack of full occlusion (as implied by the use of “though”).

Line 646: Should be “longipterygids” instead of “longipteryigds”.

Lines 648–649: Should be “odonates, neuropteridans, and lepidopterans” instead of “Odinates, Neuropterids, Lepidopterids”. (The technical forms of those names are Odonata, Neuropterida, and Lepidoptera.)

Line 650: Should be “Longipterygidae” instead of “Longipteryigdae”.

Line 656: Should be “closure” instead of “closer”.

Line 668: Should be “coleopterans and odonates” instead of “coleopterids and ondontids”. (The technical form of the former is Coleoptera.)

Line 689: “mauritianus” should be italicized.

Line 693: Should be “O’Connor” instead of “O’Conner”.

Line 699: “clades” is not necessary here.

Line 715: Should be “Longirostravines” instead of “Longirostravinines”.

Line 724: Should be “similar in” instead of “similar is”.

Line 727: Should be “Longirostravinae” instead of “Longirostravine”.

Line 730: Should be “have placed” instead of “have place”.

Line 756: Should be “Evidence of gizzard stones” instead of “Evidence gizzard stones”.

Line 759: Should be “longipterygids” instead of “longiptergyids”.

Line 783: Should be “Animalivorous” instead of “Animalivorus”, and “rostra” instead of “rostrums”.

Line 787: Should be “rectricial” instead of “rectrices”.

Line 820: A date should be added for the Beecher reference. (In the main text, it is cited as Beecher, 1962).

Lines 849–850: The title of this reference needs to be fixed; some of the words have been joined together.

Lines 1031 and 1049: The two O’Connor et al. (2011) publications cited need to be denoted as “a” and “b”.

Fig. 1: Have the photos from Wang et al. (2015) been used with permission?

Fig. 3: The last sentence of the caption should say “Taxa sampled” (or “Taxon sample”) instead of “Taxon sampled”.

Fig. 4: “Martes pennanti” should be “Pekania pennanti”, as mentioned previously, and “Caimen [sic] niger” should be “Melanosuchus niger”. I assume that the blue outlines in D represent the mean tooth morphology (like in Fig. 3), which should probably be clarified in the caption.

Fig. 5: I suggest “as seen in primarily insectivorous species” instead of “such as primarily insectivorous species” for the caption. The last sentence should say “taxa included” instead of “taxon included”.

Fig. 6: I suggest “vertebrate-eating” instead of “vertebrate prey” for the caption. The last sentence should say “taxa included” instead of “taxon included”.

Fig. 7: The caption should specify “non-avian theropods” instead of “Mesozoic theropods”, since Longipteryx is itself a Mesozoic theropod. (This should preferably also be specified in the figure itself.) The last sentence should say “taxa included” instead of “taxon included”.

Table 1: Should be “Phalanges” instead of “Phalanxes” in the top row, third column from the right. The caption appears to be missing a word in the phrase “longipterygids exhibit comparative body masses”. I suspect this is intended to be “comparatively high body masses”, though this statement would not be strictly true given that the longirostravines are estimated as having comparable body masses to typical enantiornithines. It may be apt to say that longipterygids have small pedal digit III–tibiotarsus proportions despite Longipteryx having comparatively high body mass among sampled enantiornithines.

Literature cited:
Botelho, J.F., D. Smith-Paredes, and A.O. Vargas. 2015. Altriciality and the evolution of toe orientation in birds. Evolutionary Biology 42: 502–510. doi: 10.1007/s11692-015-9334-7

Clark, A.D. and J.K. O’Connor. 2021. Exploring the ecomorphology of two Cretaceous enantiornithines with unique pedal morphology. Frontiers in Ecology and Evolution 9: 654156. doi: 10.3389/fevo.2021.654156

Höfling, E. and A. Abourachid. 2021. The skin of birds' feet: morphological adaptations of the plantar surface. Journal of Morphology 282: 88–97. doi: 10.1002/jmor.21284

Koepfli, K.-P., K.A. Deere, G.J. Slater, C. Begg, K. Begg, L. Grassman, M. Lucherini, G. Veron, and R.K. Wayne. 2008. Multigene phylogeny of the Mustelidae: resolving relationships, tempo and biogeographic history of a mammalian adaptive radiation. BMC Biology 6: 10. doi: 10.1186/1741-7007-6-10

Sato, J.J., M. Wolsan, F.J. Prevosti, G. D’Elía, C. Begg, K. Begg, T. Hosoda, K.L. Campbell, and H. Suzuki. 2012. Evolutionary and biogeographic history of weasel-like carnivorans (Musteloidea). Molecular Phylogenetics and Evolution 63: 745–757. doi: 10.1016/j.ympev.2012.02.025

---

## Round 0.2 · Major Revisions

Dear authors,

Thank you for your submission. I feel that your manuscript has substantially improved and is much more focused following the previous round of reviews. I appreciate your attention to reviewer comments. However, I still find that there are a number of issues that will need to be remedied prior to publication, including a few new analyses and some expansion of the methods. Please submit a reviewer response document, tracked changes version, and clean version of your manuscript with your revision.

Please contact me if you have any questions.

Best,

Brandon P. Hedrick, Ph.D.



Editor Comments:

A major issue that needs to be corrected is the issue of homology of measurements. You’re looking at a wide phylogenetic range of taxa and comparing mammal teeth to crocodilian teeth to fossil bird teeth needs to be better justified. This is especially the case for the geometric morphometric analyses, which will need more explanation. Be careful throughout as well to mention which tooth you’re talking about when you refer to an animal (e.g., when you talk about Noctilio, make sure you say the canine of Noctilio rather than the teeth of Noctilio).

I mentioned this in the previous iteration, but you should expand your statistical analyses throughout the paper. Previously you were largely comparing points in plots qualitatively. I appreciate you have added a Procrustes ANOVA to your stats section, but you will need to add posthoc tests as well to compare between diets (mention your R^2, F stat, etc in your results). You should also do ANOVAs with posthoc tests in your linear morphometric analyses. These would ideally be phylogenetically corrected as well using phylogenetic comparative methods as noted before. I think a general tree would suffice for this due to the large range of taxa that are analyzed rather than a published time-calibrated tree. The issue is that it seems like many of your results are related to phylogenetic signal rather than function. This was suggested by both the previous reviewer and the current reviewer.

I think it would be good for Figure 3 to have a general phylomorphospace associated with it as well since these appear to be phylogenetic groupings.

Fitting bats into a hard/soft diet dichotomy is difficult. Many bat species are generalists and bats that specialize on hard foods would not turn down a moth. Even within species, there can be geographical variability in diet. The bat canine stuff doesn’t show much from my reading and there is no in-text figure. I think it should be removed as was suggested by the previous reviewer. I also wonder how accurate the tooth measurements are given that they were taken by calipers and bat teeth are tiny. That would need to be better justified.

One caveat that you don’t mention is that there are not many vertebrate eating bats out there for comparison. I think that would need to be more fully fleshed out.

Where did diet regimes come from? Many of these species (bats are the group I know more about) are generalists and take advantage of multiple prey types. How did you account for this? This will need to be expanded in the methods.

Line by Line:

Line 146: I think it would be helpful to add sample sizes for the different groups here so readers don’t have to head to the supplement.

Line 153–155: There needs to be some statement here regarding analogy. Landmarks should theoretically be homologous, but at worst geometrically analogous. More info here is needed.

Line 164: principal

Line 169–170: Were posthoc tests done? This can be done in RRPP.

Line 182: What percent of species had two specimens measured? Male and female? Was this a mixture of sexes? Thoughts on ways to deal with sexual dimorphism in cranial length?

Line 172–184: Why not do statistics on this rather than just plot variation? You could look at bill length by dietary group like you did with shape and dietary group using a phylogenetically informed ANOVA in the phytools package in R. This applies to the bat canine measurements as well.

Line 200–202: You did this with calipers? I would imagine the variance would be extremely small since bats have tiny, tiny teeth. Measuring the width of the canine for example seems very difficult to do with calipers. What did you do to reduce error?

Line 229: micro/megachiropteran are out of date terms

Line 251: You can test this with a statistical model

Line 263–265: Many bat species eat a combination of soft and hard bodied prey. This is likely why.

Line 328: I would tone this down. The absence of evidence is not evidence.

Line 416: This is a difficult sentence to read because of differences in homology. You mean the canine of Canus lupus? Can you add in the tooth you mean in parentheses?
Line 457: As above, Microchiroptera is a colloquial term now. Perhaps you mean non-pteropodid bats?

Line 495–497: What about cats that have rostrocaudally shortened skulls, but also catch fast, small prey?

Line 530–533: Squamates should be in the next sentence with juvenile birds and amphibians. Is squamates the right word? You have an alligator I believe.

Line 541–542: This seems phylogenetically constrained.

Line 544: again, you need to be clear which tooth you’re talking about when you refer to mammals.

Lines 575–586: I think this section should be expanded. Although your data may suggest insectivory, they do not rule out vertebrate carnivory.

Fig 3: Why do the blue dots look different along the same PC? Am I reading it right that they are means?

Reviewer 2 ·

Basic reporting

This manuscript has been substantially improved and streamlined compared to the previous version. My comments on the new submission are as follows:

Lines 101–102 and 390: Wang et al. (2013) also briefly disputed mud-probing habits in longirostravines on the basis of pedal morphology.

Line 591: Botelho et al. (2015) (already cited elsewhere in the text) should be mentioned for observing a correlation between pedal ecomorphology and development.

Throughout the manuscript, there are references to the traditional division of chiropterans into Microchiroptera and Megachiroptera (e.g., Lines 191, 229, 455–460, 544, 654; Fig. S5). However, there is increasing consensus among recent phylogenetic studies that bats should instead be divided into the Yinpterochiroptera and Yangochiroptera, with some former “microchiropterans” being more closely related to megachiropterans (e.g., Teeling et al., 2005; Tsagkogeorga et al., 2013; Hawkins et al., 2019).

Literature cited
Botelho, J.F., D. Smith-Paredes, and A.O. Vargas. 2015. Altriciality and the evolution of toe orientation in birds. Evolutionary Biology 42: 502–510. doi: 10.1007/s11692-015-9334-7

Hawkins, J.A., M.E. Kaczmarek, M.A. Müller, C. Drosten, W.H. Press, and S.L. Sawyer. 2019. A metaanalysis of bat phylogenetics and positive selection based on genomes and transcriptomes from 18 species. PNAS 116: 11351–11360. doi: 10.1073/pnas.1814995116

Teeling, E.C., M.S. Springer, O. Madsen, P. Bates, S.J. O’Brien, and W.J. Murphy. 2005. A molecular phylogeny for bats illuminates biogeography and the fossil record. Science 307: 580–584. doi: 10.1126/science.1105113

Tsagkogeorga, G., J. Parker, E. Stupka, J.A. Cotton, and S.J. Rossiter. 2013. Phylogenomic analyses elucidate the evolutionary relationships of bats. Current Biology 23: 2262–2267. doi: 10.1016/j.cub.2013.09.014

Wang, X., L.M. Chiappe, F. Teng, and Q. Ji. 2013. Xinghaiornis lini (Aves: Ornithothoraces) from the Early Cretaceous of Liaoning: an example of evolutionary mosaic in early birds. Acta Geologica Sinica 87: 686–689. doi: 10.1111/1755-6724.12080

Experimental design

Although I did not flag this issue earlier, I agree with the editor and reviewer 1 from the last round of reviews that correcting for a phylogenetic signal in the data is essential to these types of analyses. Phylogenetic constraints can have far-reaching effects even across distantly related clades (e.g., conserved limb musculature across tetrapods), so I do not find the authors’ reasoning to forgo this step compelling. Furthermore, the objection that there is vast phylogenetic separation among the taxa involved does not apply to some of the analyses (e.g., the dataset of rostral proportions only includes birds). I would thus strongly echo the recommendation that phylogenetic effects should be addressed and corrected for.

Validity of the findings

No comment.

Additional comments

Line 42: Should be “Gaviiformes” instead of “Gaviformes”.

Line 85: Should be “Ichthyornis” instead of “Icththyornis”.

Line 91: Should be “Longirostravinae” instead of “Longirostavinae”.

Line 95: Crustaceans are invertebrates, so this can just read “invertebrates”.

Line 104: There does not need to be a comma between “analyses” and “to”.

Line 143: Should be “sauropsid” instead of “sauroposid”.

Line 147: Should be “included taxa” instead of “included taxon”.

Line 151: Should be “high enough resolution images to negate […] “ instead of “large enough resolution images that negated […]”.

Line 164: Should be “principal” instead of “principle”.

Line 174: Should be “variety of extant birds” instead of “variety extant birds”.

Line 228: Based on the previous description, this should be “less apicodistally curved” instead of “more apicodistally curved”.

Line 237: Should be “makes” instead of “make”.

Lines 252–253: Suggest rewording to “Other dietary preferences, such as nectivory, occupy more limited regions of the morphospace.”

Lines 255, 280, and 487: Should add a comma before “which”.

Line 293: Suggest specifying that “the closest relative of M. dentatus” refers to Schizooura.

Line 343: Suggest changing “stem avian” to “early avian”. As an extinct taxon more closely related to birds than to any other extant groups, Anchiornis would still be a stem avian even if it is a troodontid.

Line 344: Since the same sentence acknowledges that Anchiornis may be an early avian, “non-avian paravian” may not be appropriate. Maybe “stemward paravian” would be a better descriptor here.

Line 345: Should be “longipterygids” instead of “longipterytgids”.

Lines 357, 365, 372, 554, 611, and 624: Should be “longipterygines” instead of “longipteryginids”.

Line 386: “e.g.,” should be added before “genera Haematopus and Tringa”, since these are not the only genera considered shorebirds.

Line 398: Should be “Longipterygidae” instead of “Longipteryigdae”.

Line 465: Should be “predators’” instead of “predator’s”.

Line 483: Should be “predominant” instead of “predominate”.

Line 485: Should be “engulfing” or “engulfment” instead of “engulf”.

Line 503: Suggest removing “and dispatch” here, as it’s redundant with “dispatch” also used later in the same sentence.

Line 514: I would remove “the” before “L. chaoyangensis”. I also suggest specifying that the B. zhengi holotype is the only known specimen of that species.

Lines 531–533: Suggest moving the mention of squamates to the second sentence, placing them alongside the other vertebrates.

Lines 536, 549, 554, 568, 571, and 625: Should be “longirostravines” instead of “longirostravinids”.

Line 548: Should be “animalivorous” instead of “animalivorus”.

Line 555: There’s an unnecessary hyphen between “less” and “crack”.

Line 556: “both” should be removed, as three items are listed.

Line 569: Should be “furnariids” instead of “furnarids”.

Line 591: Should be “birds” instead of “bird”.

I appreciate that the authors added Bright et al. (2016) and Bright et al. (2019) to the reference list per my previous suggestions, but these studies are not mentioned in the text.

Fig. S3: “Martes pennanti” should be “Pekania pennanti” and “Caimen [sic] niger” should be “Melanosuchus niger”.

---

## Round 0.3 · Minor Revisions

Dear authors,

Thank you for your diligent work revising your manuscript. I now see it as only requiring a few minor revisions prior to publication. In addition to the reviewer's comments, I have also added a number of comments below that need to be addressed prior to publication. Please submit a response to reviewer document and tracked changes version of your manuscript and do not hesitate to contact me if you have any questions.

Best,

Brandon P. Hedrick, Ph.D.

Editor Comments:

There is not a consistent use of diet variables in the paper (also mentioned by the reviewer). Lines 148–163 set up the diet terms. However, lines 189–190 use a different set for the GM data and lines 230–231 use a third set of variables. This must be remedied prior to publication.

Additionally, more clarity in the methods employed is needed. See below in the line-by-line comments.

Line 125: ‘containing’ (to match previous wording for Yinpterochiroptera)

Line 143: Why have 8 written out when other numbers are not?

Line 146: taxon

Line 159: Myers et al., 2022 not in the references. Double check all references.

Line 187: Need open parenthesis

Line 187–190: This is confusing considering the explanation of diet definitions in the previous section. Were those diets just used qualitatively?

Line 193: Version 4.0? The version does change things in geomorph so please make note of exactly which version was used.

Line 194–197: Why do lm and pgls? Why not just pgls? Add in some justification here.

Line 203–206: Version numbers for phangorn and phytools

Line 221–223: Do you mean within species variation? Were specimens with obvious damage unincluded? This reads a bit confusingly.

Line 231: This uses a third set of diets? This is the first time nectarivory is mentioned.

Line 233: version for caper

Line 233: Does ‘ecological trait’ refer to either body mass or diet? This should be made explicit.

Line 237: Why use a Procrustes anova here rather than just a regular pgls? This isn’t multivariate shape data right? The same question for lines 242–244.

Line 239: Is this one tree or multiples? Grammar

Line 271–280: I think this should be shortened quite a bit. Also, when referring to heterodont teeth like in mammals, which tooth? Maxillary teeth is not very clear. I’m guessing canine?

Line 283: What do you mean ‘major tooth variation’? You can just remove this sentence and have the next sentence be, ‘In the PCA results, PC1 (44.01%) describes…’

Line 310: I’m guessing you did pairwise comparisons here? You should mention the RRPP package which does this rather than geomorph. This should be in the methods as well.

Line 310: For clarity, I would specify ‘Results of the non-phylogenetically corrected Procrustes ANOVA…’

Line 326: ‘surpass’

Line 438: Why italics for ‘a microraptorine’?

Line 523: Is this just carnivorans? It would be helpful to be very clear about which group you’re talking about in your references throughout the discussion because you’re looking at such a wide group of tetrapods

Line 528: ‘These data’. ‘Data’ is plural.

Line 564: How was this multivariate? Isn’t this just rostral proportion ~ body mass? Clarify here and in the methods.

Line 651: ‘that are elongate’

Line 693–695: Issues with parentheses here

Line 693–700: This altricial/precocial thing seems not super well supported. I think it would be better to remove since it is outside of the argument you are trying to make.

Line 706: Only genus/species are italicized

Reviewer 2 ·

Basic reporting

I thank the authors for their time and attention given to addressing my previous concerns. My generally minor comments on the new submission are as follows:

In this version of the manuscript, animalivory is defined as “a diet consisting of both invertebrate and vertebrate animals nearly equally” (Lines 149–150). However, the rest of the manuscript seems to use it to cover all types of diets consisting of animals (e.g., Lines 257, 439, 605, 706), regardless of the relative proportions of vertebrate to invertebrate prey. I suggest returning to the broader definition of animalivory that was originally used, as it both seems to align better with the conventional understanding of the term and seems more useful in the context of this manuscript.

Lines 258–259: Another relevant paper to cite regarding microraptorine diets came out while this manuscript was in preparation, Hone et al. (2022).

Line 456: Another relevant paper to cite regarding the lack of cranial kinesis in enantiornithines came out while this manuscript was in preparation, Wang et al. (2022).

Lines 497–499: Instead of saying only five genera of yangochiropterans forage for vertebrate prey, it may be sufficient to say that only a few are known to do so. Several examples of such have only been recently recognized, see e.g., records of bird predation in Nyctalus (Dondini and Vergari, 2000) and Chrotopterus (Perrella et al., 2020).

Literature cited
Dondini, G. and S. Vergari. 2000. Carnivory in the greater noctule bat (Nyctalus lasiopterus) in Italy. Journal of Zoology 251: 233–236. doi: 10.1111/j.1469-7998.2000.tb00606.x

Hone, D.W.E., T.A. Dececchi, C. Sullivan, X. Xu, and H.C.E. Larsson. 2022. Generalist diet of Microraptor zhaoianus included mammals. Journal of Vertebrate Paleontology advance online publication. doi: 10.1080/02724634.2022.2144337

Perrella, D.F., P.V.Q. Zima, L. Ribeiro-Silva, C.H. Biagolini Jr, A.P. Carmignotto, P.M. Galetti Jr, and M.R. Francisco. 2020. Bats as predators at the nests of tropical forest birds. Journal of Avian Biology 51: e02277. doi: 10.1111/jav.02277

Wang, M., T.A. Stidham, J.K. O'Connor, and Z. Zhou. 2022. Insight into the evolutionary assemblage of cranial kinesis from a Cretaceous bird. eLife 11: e81337. doi: 10.7554/eLife.81337

Experimental design

Although the authors indicate in their response letter that they did not sample extant bird species with marked cranial sexual dimorphism, I notice that there is at least one species in Table S4 that does: Phoeniculus purpureus, in which males have substantially longer bills than the females (Radford and du Plessis, 2004). I suspect that this is not likely to significantly affect the analytical results, but it should probably be acknowledged per the editor’s previous comments.

Literature cited
A.N. Radford and M.A. du Plessis. 2004. Extreme sexual dimorphism in Green Woodhoopoe (Phoeniculus purpureus) bill length: a case of sexual selection? The Auk 121: 178–183. doi: 10.1093/auk/121.1.178

Validity of the findings

No comment.

Additional comments

Line 115: Should be “grasping” instead of “grasp”.

Line 126: “microchiroptera” should be capitalized.

Line 143: Suggest specifying “10 non-dinosaurian reptiles” (or “non-dinosaurian sauropsids”).

Line 144: Should be “mass and probable diet […] were included” instead of “was included”.

Line 146: Should be “taxa of similar size […] were used” instead of “was used”.

Line 161: Should be “is nectarivory” instead of “being nectarivory”.

Lines 288 and 289: Since the teeth were landmarked in lateral view, should this be “mesiodistally” instead of “buccolingually”?

Line 374: Should be “Longipteryginae” instead of “Longiptergyinae”.

Line 376: Should be “the latter of which” instead of “the latter which”.

Line 436: I think this is meant to read “longipterygids are similar in size to […]”, otherwise the subject of the sentence is ambiguous.

Line 438: “a microraptorine” shouldn’t be italicized.

Lines 517–518: Suggest specifying “process (i.e., chew) food in the jaws […]”.

Line 583: Suggest “the B. zhengi holotype” instead of just “B. zhengi specimen”.

Line 600: Birds are sauropsids, so this can read “juvenile sauropsids and amphibians” or “juvenile birds, non-avian sauropsids, and amphibians”.

Line 612: Should be “maxillary teeth” instead of just “maxillary”.

Line 636: Should be “rostrum length” instead of just “rostrum”.

Lines 651–652: Should be “that elongate distally” instead of “the elongate distally”.

Line 689: Should be “ Ramphastidae” instead of “Rhamphastidae”.

Line 706: “microraptorines” shouldn’t be italicized.

Line 736: Should be “provide support” instead of “provides support”.

Line 739: Should be “give us insight” instead of “gives us insight”.

Line 742: “feather impressions” may be more appropriate than “rectricial impressions” here, since the discussion on rectrices has been removed from this version.

Caption to Fig. 2: Should be “a longipterygid” instead of “alongipterygid”.

---

## Round 0.4 · Minor Revisions

Dear authors,

Thank you for your revision. I have a few points that should be addressed before publication. Be sure to respond to them in the response to reviewers document so I know what changes are made in addition to changing them in the tracked changes version of the manuscript.

The dietary inconsistencies must be addressed prior to publication. On lines 147–162, you mention animalivory, carnivory, piscivory, invertivory, insectivory, herbivory, nectarivory, and omnivory (8 categories). On lines 186–189, you limit this to 5 categories. Why are there multiple classifications here? Did you not have any examples of animalivores, invertivores, or nectarivores in your dataset so you only have five categories that you code as binary categorical variables? The dietary categorization from lines 232–241 still has a separate subset. You only mention 7 categories here (skipping animalivory). I think you should put all of your diet categories into the same paragraph and if you do subset them, refer to them as perhaps ‘full complement of diets’, ‘diet subset 1’. and ‘diet subset 2’ or something along those lines. You should also explain why it was necessary to subset. As of now, it is not clear which were used or why.

Line 144: ‘data were’. ‘data’ is plural. Check this throughout.

Line 218–220: I see that this was added in response to reviewer comments. You might mention something more general about sexual dimorphism not being considered and Phoeniculus being an example. I agree with the reviewer that this is unlikely to cause problems for your analyses.

Line 277: which maxillary teeth? Canines?

Line 337: The ‘this suggests…’ in this paragraph should be moved to the discussion. This is all discussion rather than results. I would also say that none of these diets can be ruled out, but you could make an argument for them being less likely based on your results.

Line 444: Although you made microraptorine non-italics, the a in this sentence is still italicized


Thank you for your submission to PeerJ. Let me know if you have any additional questions.

Best,

Brandon P. Hedrick, Ph.D.

---

## Round 0.5 · accepted · Accept

Dear authors,

Thank you for your revisions. I had only mentioned one instance in the last paragraph of your results concerning ruling out hypotheses. However, this appeared twice (also on line 358-359) and I think this should be removed prior to publication.

Thank you for your submission to PeerJ.

Best,

Brandon P. Hedrick